# On Slicing Optimality for Mutual Information

**Ammar Fayad**[*]
MIT

**Majd Ibrahim**
HIAST

## Abstract

Measuring dependence between two random variables is of great importance in various domains but is difficult to compute in today's complex environments with high-dimensional data. Recently, slicing methods have shown to be a scalable approach to measuring mutual information (MI) between high-dimensional variables by projecting these variables into one-dimensional spaces. Unfortunately, these methods use uniform distributions of slicing directions, which generally discard informative features between variables and thus lead to inaccurate quantification of dependence. In this paper, we propose a principled framework that searches for an *optimal* distribution of slices for MI. Importantly, we answer theoretical questions about finding the optimal slicing distribution in the context of MI and develop corresponding theoretical analyses. We also develop a practical algorithm, connecting our theoretical results with modern machine learning frameworks. Through comprehensive experiments in benchmark domains, we demonstrate significant gains in our information measure than state-of-the-art baselines.

## 1  Introduction

Mutual information (MI) measures statistical dependence between two random variables by quantifying the amount of information gained about one variable from an observation of the other variable (Shannon, 1948). Despite its popularity in various fields (Marinoni and Gamba, 2017; Hamma et al., 2016), MI suffers from the curse of dimensionality: the estimation of mutual dependence is often stymied by the large dimension of the variables (Kraskov et al., 2004). Recently, slicing methods (Bonneel et al., 2015; Nadjahi et al., 2020) have demonstrated a scalable approach for estimating probability divergences and mutual dependence. Specifically, sliced mutual information ($\mathcal{SI}$; Goldfeld and Greenewald (2021)) measures the average of MI terms between one-dimensional projections of the high-dimensional variables, where the projecting slicing directions are sampled uniformly from a unit sphere. Because slicing methods do not compute the dependence directly in the high-dimensional space but in the low-dimensional space, these methods significantly reduce computation costs and are scalable. However, a critical problem of these slicing frameworks is that they generally require many slices for accurate estimation. As we detail later in the paper, the root cause of this problem is the uniform sampling of the slicing directions: discarding informative features about the variables and generating noisy slices by equally favoring important and irrelevant regions of the variables' spaces. Hence, the uniform sampling fails to capture complex relationships of variables which renders current slicing methods inefficient and can even lead to inaccurate dependence quantification.

**Our contribution.**   With this insight, this paper aims to address the uniform sampling issue with the slicing approaches for improving the dependence measurement. Specifically, motivated by the success of Nguyen et al. (2021) in the generative modeling domain, we seek to find an *optimal* distribution of slicing directions in the context of MI that satisfies two criteria: 1) slices are distributed over maximally informative regions, and 2) slices are scattered over the unit sphere so that no

---

[*]`afayad@mit.edu`

37th Conference on Neural Information Processing Systems (NeurIPS 2023).

relevant directions are discarded. We formalize these criteria in a regularized optimization problem and introduce a novel measure of dependence, the optimal sliced mutual information (denoted by $\mathcal{SI}^*$). $\mathcal{SI}^*$ provides a much more efficient performance compared to previous slicing methods and dependence measures (e.g., improvements of up to $30\%$ in dependence detection power). We show an important benefit of $\mathcal{SI}^*$, which is that it can transform the fundamental work of Goldfeld and Greenewald (2021) (i.e. $\mathcal{SI}$) to larger and more complex machine learning problems.

Several unique challenges arise when developing our regularized optimization problem. First, MI requires at least two slicing variables and the optimization should be over the space of the joint distribution of the slices, but the constraint is enforced over the marginals. We find that this challenge essentially corresponds to solving optimal transportation (OT) (Villani, 2009) problems: optimization over a set of couplings (of the distributions of slices) with given constraints. To that end, we leverage OT properties that enable us to derive many interesting theorems about $\mathcal{SI}^*$ connecting it to MI, SI, and differential entropy. The second challenge is the continuity problem: the cost function in OT is assumed to be upper semicontinuous in the topology of weak convergence for the *maximizing* solution to exist, but this continuity assumption no longer holds when dealing with information-theoretic quantities. We rigorously prove that, under very mild assumptions, a maximizing solution exists for any input variables regardless of their dimensions.

In summary, this paper bridges the gap between intractable information/dependence measures and modern computational frameworks by addressing the shortcomings of current slicing approaches. We investigate the optimality of slicing directions for MI with the following key contributions:

- **Formalization of a new dependence measure (Sections 3.1 and 3.2).** We introduce a novel and scalable dependence measure, $\mathcal{SI}^*$ with theoretical analyses of its properties and implications.
- **Scalable estimator (Sections 3.3 and 4).** We construct an optimal estimator with the tightest bounds and explicitly show that the effect of the variables' dimensions on the convergence rate is only up to a constant factor. We then employ deep neural networks to this optimal estimator to acquire an end-to-end $\mathcal{SI}^*$ neural estimator and show that it is computationally efficient.
- **Comprehensive evaluation of our approach (Section 6).** We demonstrate significant gains of $\mathcal{SI}^*$ in the accuracy of detecting complicated relations over state-of-the-art baselines on an extensive set of experiments. $\mathcal{SI}^*$ also works excellently across challenging domains (e.g., representation and reinforcement learning), which is an unprecedented advantage and a feature that is largely missing from current literature on statistical dependence.

## 2 Background

**Notation.** We follow standard notations from Villani (2009). Specifically, this paper considers Borel measures on Polish spaces; the latter are complete, separable metric spaces, equipped with their Borel $\sigma$-algebra. We denote the space of Borel probability measures on $\mathcal{X}$ as $\mathcal{P}(\mathcal{X})$. If $\mu$ is a Borel measure on $\mathcal{X}$, and $T$ is a Borel map $\mathcal{X} \to \mathcal{Y}$, then $T_{\#}\mu$ stands for the push-forward measure of $\mu$ by $T$: it is a Borel measure on $\mathcal{Y}$, defined by $T_{\#}\mu[A] = \mu[T^{-1}(A)]$ for any $A \subset \mathcal{Y}$, where $\delta_x$ is the Dirac delta function at $x$. If $\pi(d\theta d\varphi)$ is a probability measure in two variables $\theta \in \mathcal{X}$ and $\varphi \in \mathcal{Y}$, its marginal (or projection) on $\mathcal{X}$ (resp. $\mathcal{Y}$) is the measure $p_{\#}^{\Theta}\pi$ (resp. $p_{\#}^{\Phi}\pi$), where $p^{\Theta}(\theta, \varphi) = \theta$ and $p^{\Phi}(\theta, \varphi) = \varphi$. We focus on absolutely continuous random variables with bounded density functions. Let $\theta^* : x \mapsto \theta^{\top}x$ and $\boldsymbol{\gamma}(\mathcal{X})$ is the uniform distribution on $\mathcal{X}$. For any two probability measures $\mu, \nu$, $d\mu/d\nu$ denotes the Radon-Nikodym derivative of $\mu$ with respect to $\nu$. Furthermore, let $(\psi \times \xi)(x, y) = (\psi(x), \xi(y))$ be the Cartesian product of two functions $\psi, \xi$. Denote by $\mathcal{C}(\mathcal{X}, \mathcal{Y})$ the set of continuous Borel maps from $\mathcal{X}$ to $\mathcal{Y}$. $\|.\|$ denotes the Euclidean norm. Finally, $\mathbb{S}^{d-1}$ is the unit sphere in $\mathbb{R}^d$, and $\Omega_{d,k} \overset{\text{def}}{=} \mathbb{S}^{d-1} \times \mathbb{S}^{k-1}$.

**Information theory.** The mutual information (Shannon, 1948) characterizes the distance between the joint distribution $P_{X,Y}$ and the product of its marginals $P_X \otimes P_Y$:

$$\mathcal{I}(X;Y) = \text{KL}(P_{X,Y}||P_X \otimes P_Y)) = \int_{\mathcal{X} \times \mathcal{Y}} \log\left(\frac{dP_{X,Y}}{dP_X \otimes P_Y}\right)dP_{X,Y},$$

where $\text{KL}(\cdot||\cdot)$ is the Kullback–Leibler divergence (Kullback and Leibler, 1951) which is lower semicontinuous in the topology of weak convergence (Posner, 1975). The *Sliced* mutual information (Goldfeld and Greenewald, 2021) is the mean of MI terms between one-dimensional projections of

the variables:

$$\mathcal{SI}(X;Y) = \oint_{\Omega_{d_x,d_y}} \mathcal{I}(\theta^\top X; \varphi^\top Y) d\boldsymbol{\gamma}(\theta) \otimes \boldsymbol{\gamma}(\varphi). \tag{1}$$

In Equation (1), $X \in \mathbb{R}^{d_x}, Y \in \mathbb{R}^{d_y}$; $\theta$ and $\varphi$ are the slices corresponding to $X$ and $Y$, respectively. We note that slices are independently and uniformly sampled from $\mathbb{S}^{d_x-1}$ and $\mathbb{S}^{d_y-1}$. This uniformity causes redundancy in slices which deems optimality beneficial and necessary for the slicing process.

## 3 Definition and Theories of Slicing Optimality

In this section, we begin by motivating the definition of our novel measure by showing the advantage of our solution over uniform sampling. We then formalize our proposal as a regularized optimization problem. Lastly, we discuss our theoretical findings, such as the existence of an optimal slicing policy and the properties of $\mathcal{SI}^*$. These studies show that $\mathcal{SI}^*$ possesses desired characteristics of a dependence measure.

### 3.1 Definition of $\mathcal{SI}^*$

We first motivate our solution in an example, in which we incorporate the following two optimality criteria into $\mathcal{SI}$:

1. The projection directions are mainly concentrated into areas where the one-dimensional variables contain the maximum mutual information possible.

2. The slicing directions are also diversified over the whole sphere, ensuring that all regions with relevant information are visited.

**Motivating example.** Let $Z \sim \mathcal{N}(0, I_3), \Lambda \sim \mathcal{N}(0, 0.1 I_2)$, and let $X = [Z_1, Z_2]^\top, Y = [Z_1, Z_3]^\top + \Lambda$. In this example, we aim to estimate the average MI between one-dimensional projections of the two variables $X$ and $Y$ and show that our optimal distribution of slices yields superior performance over standard $\mathcal{SI}$. Specifically, by sampling slicing directions, $\theta$ and $\varphi$, uniformly and independently from the 2-sphere (i.e., unit circle), we obtain $\mathcal{SI}$. To construct the distribution of slices that meets the former criteria, we first note that the slices which project only the first entries of $X, Y$

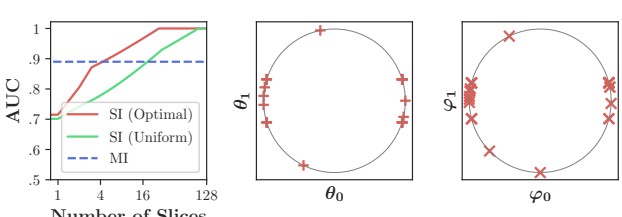

Figure 1: ROC AUC curve (left) and visualization of the custom distribution the slices (middle and right). Optimal distribution of slices can capture more information and yields a more accurate measure of dependence.

yield the maximum information because both variables share $Z_1$ as their first entry (up to additive noise) while their second entries are independent. We leverage this observation and sample two-dimensional slices such that they are mainly concentrated around $|\theta_0| = 1$, and $|\varphi_0| = 1$, respectively. We also enforce the second criterion of slices diversity by choosing a few slices away from the cluster (See Figure 1 middle and right). We refer to Appendix D.1 for details.

Figure 1 (left) shows the area under the curve (AUC) of the receiver operating characteristic (ROC) as a function of the number of slices along with a visualization of the customized distribution of $\theta, \varphi$. Note the improvement in the number of slices needed until reaching an ROC AUC = 1, as our customized measure requires $\sim 75\%$ fewer slices to reach perfect accuracy. This result shows that our proposed distribution can extract sufficient information more effectively.

Having established the importance of optimal slicing, we now formally define the optimal sliced mutual information ($\mathcal{SI}^*$) as a weighted average of information stored in one-dimensional projections of two random variables $X, Y$:

**Definition 1** (Optimal sliced mutual information). *Given two random variables $X \in \mathbb{R}^{d_x}, Y \in \mathbb{R}^{d_y}$ and $\omega_X, \omega_Y \in [0, \pi/2]$, define the following collection of probability measures $\Sigma_{d,\omega} = \{\mu : \mu \in$*

$\mathcal{P}(\mathbb{S}^{d-1})$, $\mathbb{E}_{x,y\sim\mu}[\arccos|x^\top y|] \geq \omega\}$. *The Optimal Sliced Mutual Information can be expressed as*:

$$\mathcal{SI}^*(X;Y) = \sup\left\{ \oint_{\Omega_{d_x,d_y}} \mathcal{I}(\theta^\top X; \varphi^\top Y)d\sigma(\theta,\varphi) \quad : p_\#^\Theta\sigma \in \Sigma_{d_x,\omega_X}, p_\#^\Phi\sigma \in \Sigma_{d_y,\omega_Y} \right\}. \quad (2)$$

We call the distribution $\sigma$ a slicing policy. Note that $p_\#^\Theta\sigma$ refers to the first marginal distribution of $\sigma$ (in other words, the law of the random slicing vector of $X$, $\Theta$, i.e. $\text{law}(\Theta) = p_\#^\Theta\sigma$); $\text{law}(\Theta) \in \Sigma_{d_x,\omega_X}$. This observation also applies to $\text{law}(\Phi) \in \Sigma_{d_y,\omega_Y}$. The supremum is taken to ensure that the slices convey sufficient information to quantify the relationship between $X$ and $Y$, or equivalently to ensure that the slices are prevalent in maximally informative regions (i.e., criterion 1). The set over which the supremum is taken, thus, refers to the joint measures whose marginals produce slices that are scattered over the unit sphere of an appropriate dimension (i.e., criterion 2), where we use the $\arccos|.|$ to measure slices diversity. Importantly, without the constraint, the measure might collapse to a Dirac delta probability measure at the *max* slicing vector (i.e. collapse to $\sup_{\theta,\varphi}\mathcal{I}$). As such, the prominent feature of $\mathcal{SI}^*$ is that it reveals the maximum amount of information content stored in $X, Y$ by searching for a distribution of slices that results in informative, noiseless, and diverse projections.

**Remark 1.** $\mathcal{SI}^*$ *is not a mutual information estimator and should not be used as a proxy of MI but rather as a new dependency measure. In fact, although it shares many properties with MI, $\mathcal{SI}^*$ has multiple advantages such as scalability and efficiency, not to mention the cutting-edge performance on detecting complex relationships between high-dimensional random variables in nontrivial settings.*

**Remark 2.** *If $d_x = d_y = 1$, $\mathcal{SI}^*(X;Y) = \mathcal{I}(X;Y)$. Essentially, for one-dimensional r.v.s, $\mathcal{SI}^*$ boils down to MI.*

We also consider the optimality of slicing into $k$-dimensional subspaces by enforcing our same criteria on distributions over the Stiefel manifold (Chikuse, 2012). Appendix E details this definition.

### 3.2 Theoretical Properties of $\mathcal{SI}^*$

We propose optimizing the slicing distribution towards one-dimensional projections with higher MI values. However, due to the lower semicontinuity of MI in the maximization problem, it is difficult to establish whether an optimal slicing policy exists. In the following, we prove the existence of a solution that our problem can converge to by assuming that for a given random variable $X$ with pdf $p_X$, $\int \|x\|^\kappa p_X(x)dx < \infty$ for some $\kappa > 1$:

**Theorem 1** (Existence of an optimal slicing policy). *For any random variables $X$ and $Y$ and any $\omega_X, \omega_Y \in [0, \pi/2]$, there exists a slicing policy $\sigma$ such that the information functional, $\oint \mathcal{I}(\theta^\top X; \varphi^\top Y)d\sigma(\theta,\varphi)$, is maximized among all possible couplings of $(law(\Theta), law(\Phi)) \in \Sigma_{d_x,\omega_X} \times \Sigma_{d_y,\omega_Y}$.*

*Proof.* See Appendix A.1 for details. □

We refer to such $\sigma$ as an optimal slicing policy. Theorem 1 enables us to treat the problem as a reward-maximizing OT. Roughly speaking, $\mathcal{SI}^*(X;Y)$ can be deemed as a reward-maximizing optimal transport problem, where the reward induced by slicing vectors $\theta$ and $\varphi$ is the amount of information they reveal about $X$ and $Y$. As such, the formula in Equation (2) measures the maximum information gain possible among slicing policies.

Rényi (1959) and Bell (1962) postulate that a measure of dependence should satisfy properties found in Theorem 2:

**Theorem 2** (Properties of $\mathcal{SI}^*$). *For random variables $X$ and $Y$, we have:*

1. $\mathcal{SI}^*(X;Y)$ *is nonnegative and symmetric.*

2. $\mathcal{SI}^*(X;Y) = 0$ *if and only if $X$ and $Y$ are independent.*

3. *If $X_n, Y_n$ are sequences of random variables with joint distribution $P_{X,Y}^{(n)}$ that converges pointwise to the joint distribution $P_{X,Y}$, then $\lim_{n\to\infty}\mathcal{SI}^*(X_n;Y_n) = \mathcal{SI}^*(X;Y)$.*

4. *Similar to MI, $\mathcal{SI}^*$ has a relative entropy form:* $\mathcal{SI}^*(X;Y) = \sup_\sigma \mathbb{E}_{(\Theta,\Phi)\sim\sigma}\Big[KL\big((\Theta^* \times \Phi^*)_\#P_{X,Y}||\Theta_\#^*P_X \otimes \Phi_\#^*P_Y\big)\Big] = \sup_\sigma KL\big(\sigma \otimes (\Theta^* \times \Phi^*)_\#P_{X,Y}||\sigma \otimes \Theta_\#^*P_X \otimes \Phi_\#^*P_Y\big).$

*Proof.* See Appendix A.2 for details. □

Now that we have established that $\mathcal{SI}^*$ possesses the desirable properties of a dependence measure, we emphasize the importance of using two slices:

**Remark 3.** *Unlike existing slicing methods (Nguyen et al., 2021), $\mathcal{SI}^*$ requires two slicing variables $\Theta$ and $\Phi$ since using only one slice violates the important property (2) in Theorem 2. To illustrate, Let $X \sim \mathcal{N}(0, I_2)$ and $Y = \begin{pmatrix} 0 & \alpha \\ -\alpha & 0 \end{pmatrix} X, \alpha \neq 0$, clearly $X$ and $Y$ are dependent but $\theta^\top X$ and $\theta^\top Y$ are not for all $\theta \in \mathbb{S}^1$ (the latter follows from the independence of the entries of $X$. See (Goldfeld and Greenewald, 2021)). The observation implies that for the such dependent $X, Y, \mathcal{SI}^*(X; Y) = 0$. We thus resort to using two slicing directions.*

Next, we use the property (4) in Theorem 2 to represent $\mathcal{SI}^*$ using a discriminator function.

**Corollary 1** (Discriminator-based form). *Let $\mu = (\Theta^* \times \Phi^*)_\# P_{X,Y}, \nu = \Theta^*_\# P_X \otimes \Phi^*_\# P_Y$. We write $\mathcal{SI}^*$ as:*

$$\mathcal{SI}^*(X; Y) = \sup_\sigma \mathbb{E}_{(\Theta, \Phi)} \mathbb{E}_\mu \Big[ \arg\max_h \big[ \mathbb{E}_\mu[\log(\varsigma \circ h(.))] - \mathbb{E}_\nu[1 - \log(\varsigma \circ h(.))] \big] \Big],$$

*where $\varsigma$ is the Sigmoid function and the discriminator $h$ is defined on $\mathbb{R}$.* This GAN-type form may be used to estimate $\mathcal{SI}^*$ in Reproducing Kernel Hilbert Space similarly to Ghimire et al. (2021)'s approach for estimating KL.

*Proof.* See Appendix A.2.1 for details. □

Moreover, we present a variational representation of $\mathcal{SI}^*$ which will be later used to construct a neural estimator:

**Corollary 2** (Variational representation). *Let $(X, Y) \sim P_{X,Y}$ and $(\Psi, \Upsilon) \sim \boldsymbol{\gamma}(\Omega_{d_x, d_y})$. We have:*

$$
\begin{aligned}
\mathcal{SI}^*(X; Y) = \sup_{T, f_1, f_2} \Big\{ &\mathbb{E}\big[ T(\Psi, \Upsilon, f_1(\Psi, \Upsilon)^\top X, f_2(\Psi, \Upsilon)^\top Y) \big] - \\
&\log \mathbb{E}\big[ \exp\big( T(\Psi, \Upsilon, f_1(\Psi, \Upsilon)^\top \overline{X}, f_2(\Psi, \Upsilon)^\top \overline{Y}) \big) \big] \Big\},
\end{aligned}
\tag{3}
$$

*where $(\overline{X}, \overline{Y}) \sim P_X \otimes P_Y$. The supremum is taken over $T \in \mathcal{C}(\Omega_{d_x, d_y} \times \mathbb{R}^2, \mathbb{R}), f_1 \in \mathcal{F}_{\omega_X} \stackrel{def}{=} \{f : f \in \mathcal{C}(\Omega_{d_x, d_y}, \mathbb{S}^{d_x - 1}), f_\# \boldsymbol{\gamma}(\Omega_{d_x, d_y}) \in \Sigma_{d_x, \omega_X}\}, f_2 \in \mathcal{F}_{\omega_Y}$, where $\mathcal{F}_{\omega_Y}$ is defined analogously to $\mathcal{F}_{\omega_X}$.*

*Proof.* See Appendix A.2.2 for details. □

Corollaries 1 and 2 provide useful representations of $\mathcal{SI}^*$ that make it compatible with modern machine learning algorithms and optimization models. Next, we highlight the concept of data processing that connects $\mathcal{SI}^*$ to $\mathcal{I}$ and $\mathcal{SI}$.

**Theorem 3** (Data processing). *For $\omega_X, \omega_Y \geq \pi/4$,*

$$\mathcal{I}(X; Y) \geq \mathcal{SI}^*(X; Y) \geq \mathcal{SI}(X; Y). \tag{4}$$

*Notably, $\mathcal{I}(X; Y) \geq \sup_{\theta, \varphi} \mathcal{I}(\theta^\top X; \varphi^\top Y) \geq \mathcal{SI}^*(X; Y) \geq \mathcal{SI}(X; Y) \geq \inf_{\theta, \varphi} \mathcal{I}(\theta^\top X; \varphi^\top Y)$.*

*Proof.* See Appendix A.3 for details. □

Theorem 3 shows that $\mathcal{SI}^*$ is a generalization of $\mathcal{SI}$ because, for the given values of $\omega_X$ and $\omega_Y$, $\Sigma_{d_x, \omega_X}$ and $\Sigma_{d_y, \omega_Y}$ contain the uniform distributions over the respective sphere. On the other hand, $\mathcal{SI}^*$ does not exceed MI as a consequence of the Data Processing Inequality. However, due to the scalability and sample efficiency issues in MI, $\mathcal{SI}^*$ is a much better candidate for quantifying dependence than MI especially when the data have complex structures.

### 3.3 Estimation

A key property of $\mathcal{SI}^*$ is its scalability which is intrinsic to slicing approaches, where the latter facilitate estimation from samples and do not suffer from the curse of dimensionality since computations are performed in low-dimensional subspaces. Here, we construct an optimal $\mathcal{SI}^*$ estimator based on

one-dimensional information estimators. Let $\widehat{\mathcal{I}}_n$ be a one-dimensional MI estimator over $n$ samples whose absolute error is uniformly bounded by $\delta(n)$:

$$\sup_{P_{S,Q}} \mathbb{E}[|\mathcal{I}(S;Q) - \widehat{\mathcal{I}}_n(S;Q)|] \leq \delta(n).$$

Any optimal slicing policy $\sigma^*$ that maximizes the information functional can be obtained by applying an appropriate transformation on the uniform measure on $\Omega_{d_x,d_y}$; this transformation can be learned using neural networks. By the definition of the push-forward measure, we express $\mathcal{SI}^*(X;Y)$ as[2]:

$$\mathcal{SI}^*(X;Y) = \sup_{f_1 \in \mathcal{F}_{\omega_X}, f_2 \in \mathcal{F}_{\omega_Y}} \mathbb{E}_{(\psi,\upsilon) \sim \boldsymbol{\gamma}(\Omega_{d_x,d_y})} \big[ \mathcal{I}(f_1(\psi,\upsilon)^\top X; f_2(\psi,\upsilon)^\top Y) \big],$$

where $\mathcal{F}_{\omega_X}, \mathcal{F}_{\omega_Y}$ are defined in Corollary 2. The approximation then becomes:

$$\widehat{\mathcal{SI}}^*_{n,m}(X;Y) \stackrel{\text{def}}{=} \sup_{f_1 \in \mathcal{F}_{\omega_X}, f_2 \in \mathcal{F}_{\omega_Y}} \frac{1}{m} \sum_{j=1}^m \big[ \widehat{\mathcal{I}}_n(f_1(\psi_j,\upsilon_j)^\top X; f_2(\psi_j,\upsilon_j)^\top Y) \big]. \tag{5}$$

**Theorem 4** (Convergence Rate). *The uniform error bound of $\widehat{\mathcal{SI}}^*_{n,m}(X;Y)$ is:*

$$\sup_{P_{X,Y}} \mathbb{E}[|\mathcal{SI}^*(X;Y) - \widehat{\mathcal{SI}}^*_{n,m}(X;Y)|] \leq \delta(n) + \frac{U}{2\sqrt{m}},$$

*where $U \propto (d_x^{-1} + d_y^{-1})^{1/2}$ is a constant factor.*

*Proof.* See Appendix A.4 for details. $\qquad\square$

We note that the estimation rate depends on the dimensions of the problem $d_x, d_y$ only up to the constant factor $U$. The explicit dependence of $U$ on $d_x, d_y$ has been a recent challenge which is now solved in Theorem 4. By imposing additional regularity on $P_{X,Y}$, we can obtain $\delta(n) \leq C\big((n\log n)^{-\frac{s}{s+2}} \log n^{1-\frac{2}{p(s+2)}} + n^{-1/2}\big)$ where $s,p : 0 < s \leq 2 \leq p$, and $C > 0$ (see Han et al. (2020); Goldfeld and Greenewald (2021)), which is significantly faster than the MI estimation rate, i.e. $n^{-1/(d_x+d_y)}$.

# 4 Neural Estimator for $\mathcal{SI}^*$

In this section, we introduce a practical implementation of $\mathcal{SI}^*$ (based on Corollary 2) by leveraging a neural network estimator and connect our theoretical results in Section 3 with modern machine learning frameworks. Let $\mathcal{D} \stackrel{\text{def}}{=} \{(X_k, Y_k)\}_{k=1}^m$ be a mini-batch with $X \in \mathbb{R}^{d_x}, Y \in \mathbb{R}^{d_y}$, and $n = \lfloor \frac{m}{2} \rfloor$. We take $(X,Y) \sim \{(X_k, Y_k)\}_{k=1}^n$, $(\overline{X}, \overline{Y}) \sim \{(X_k, Y_{k+n})\}_{k=1}^n$,[3] and $(\Psi_k, \Upsilon_k) \sim \boldsymbol{\gamma}(\Omega_{d_x,d_y})$. Then, the neural network estimator $\mathcal{SI}^*_{\mathbb{W}}$ is computed as:

$$\sup_{T,f_1,f_2} \left\{ \frac{1}{n} \sum_{k=1}^n T(\Psi_k, \Upsilon_k, f_1^{(k)\top} X_k, f_2^{(k)\top} Y_k) - \log \frac{1}{n} \sum_{k=1}^n \exp\left( T(\Psi_k, \Upsilon_k, f_1^{(k)\top} \overline{X}_k, f_2^{(k)\top} \overline{Y}_k) \right) \right.$$
$$\left. + \lambda_1 \left( \frac{1}{n^2} \sum_{k,j} \arccos|f_1^{(k)\top} f_1^{(j)}| - \omega_X \right) + \lambda_2 \left( \frac{1}{n^2} \sum_{k,j} \arccos|f_2^{(k)\top} f_2^{(j)}| - \omega_Y \right) \right\}, \tag{6}$$

where $f_i^{(k)} = f_i(\Psi_k, \Upsilon_k)$ for $i = 1,2; k \in [n]$. $\lambda_1, \lambda_2 > 0$, and $\mathbb{W} \stackrel{\text{def}}{=} [\boldsymbol{w}_T, \boldsymbol{w}_1, \boldsymbol{w}_2]$ are the parameters of neural networks $T, f_1, f_2$, respectively. We also follow Song and Ermon (2020) and account for the possible high-variance issues by using a smoothed version of Equation (6), where we clip the $\exp(\cdot)$ term in the second line between $\exp(-\rho)$ and $\exp(\rho)$ where $\rho > 0$. We refer to Appendix C for the pseudocode of $\mathcal{SI}^*_{\mathbb{W}}$. We later analyze the computational complexity of the estimator.

---

[2]See detailed mathematical explanation in our proof of Corollary 2.
[3]Alternatively, one can take $(X,Y) \sim \{(X_k, Y_k)\}_{k=1}^n$, and $(\overline{X}, \overline{Y}) \sim \{(X_k, Y_{\sigma(k)})\}_{k=1}^n, \sigma(k) \neq k, \sigma \in \mathcal{S}$ (the group of $n$-permutations).

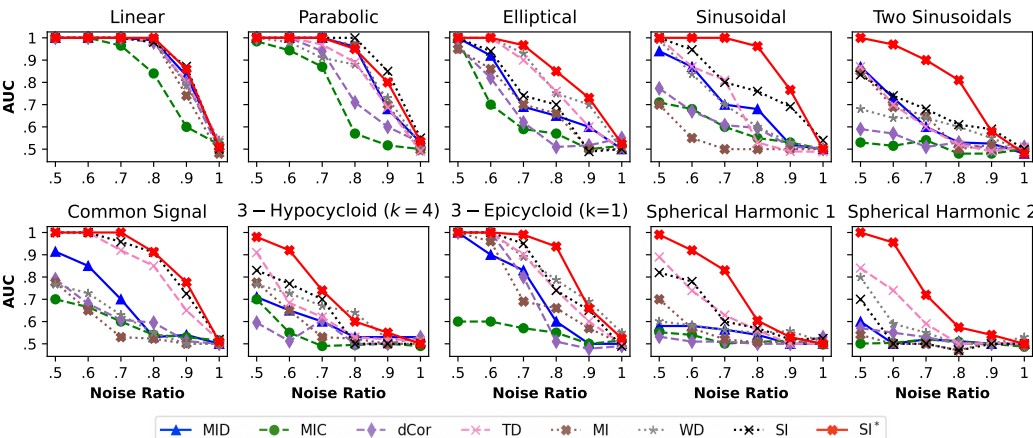

Figure 2: Statistical efficiency of dependence measures with structures of varying complexity $X \in \mathbb{R}^3, Y \in \mathbb{R}^3$. The AUC ROC quantifies the measure's discriminative ability, where a perfect classifier has AUC ROC of 1.0 while a random classifier has AUC ROC of 0.5. The figures are computed from 100 random runs.

## 5 Related Work

The main idea in quantifying dependence between random variables $X, Y$ is to characterize the distance between their joint distribution $P_{X,Y}$ and the product of its marginals $P_X \otimes P_Y$. Naturally, this problem has been studied in the area of optimal transportation theory (Mordant and Segers, 2022; Wiesel, 2021; Chuang et al., 2023). For example, Nies et al. (2021) defined the distance as the optimal transport cost on the spaces of the aforementioned distributions for a restricted set of cost functions. Other popular approaches study the distance covariance, which is the weighted Euclidean distance between the joint characteristic function and the product of the marginals' characteristic functions (Székely et al., 2007). Also, Reshef et al. (2011) introduced MIC, a measure of dependence for the 2-variable relationships based on testing multiple grids with varying dimensions and searching for the grid with maximum information between the two variables. Notably, (Liu et al., 2021) benefit from the optimal transportation theory to calculate Squared-loss Mutual Information with a small number of paired samples (drawn from the joint distribution) and a large number of unpaired samples. Lastly, MID (Sugiyama and Borgwardt, 2013) uses the concept of fractal dimensions (Ott, 2002) to define the information dimension and shows benefits in various applications.

In the context of sliced statistical distances, Nguyen et al. (2021) proposed a similar solution to alleviate the problem of using a large number of slices and applied it to the Wasserstein (W-) distance by reformulating the distance as an optimization problem. While we share a similar motivation, the two slicing approaches have significant differences stemming from the fact that Sliced W-distances have only been studied in the context of generative modeling; they only require one slicing direction; and their cost function (1-Wasserstein distance) is continuous, which makes their problem a vanilla expectation maximization (EM) problem. In a different domain, Nguyen et al. (2020) proposed an improvement to relational regularized autoencoders by finding an important area of projections characterized by a von Mises-Fisher (vMF) distribution (Jupp and Mardia, 1979). Optimizing over the family of vMF distributions helps to identify important directions on the sphere ($\epsilon$ parameter) and how much weight to put there compared to other directions ($\kappa$ parameter). We, however, do not restrict our optimization to a particular family of distributions since the optimal slicing policy might not take the form of vMF.

## 6 Experiments

We conduct extensive experiments to demonstrate $\mathcal{SI}^*$'s efficacy. We compare $\mathcal{SI}^*$'s dependence quantification quality to a diverse set of competitive baselines. Through careful analyses, we show $\mathcal{SI}^*$'s scalability, computation speed, sample efficiency, and slicing efficiency. We further provide empirical evidence that $\mathcal{SI}^*$ works excellently across challenging representation learning and rein-

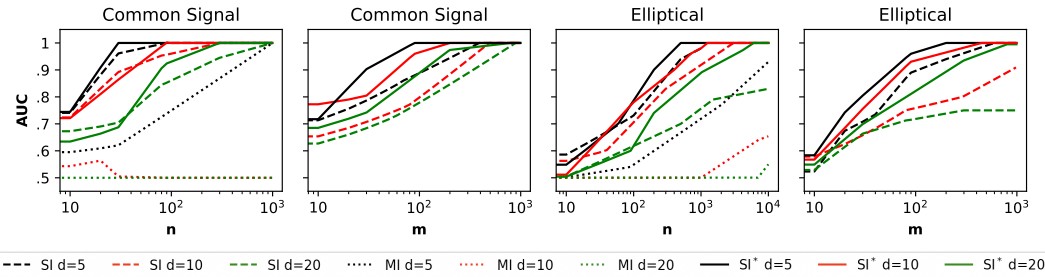

Figure 3: Statistical efficiency of $\mathcal{SI}^*$, SI, and MI with dimension, number of samples ($n$), and number of slices ($m$). The figures with respect to $m$ do not include MI because this method does not use slices.

forcement learning tasks. We refer to Appendix B for further analysis experiments and Appendix D for implementation details and hyperparameters provided at `https://bit.ly/3foLke2`.

## 6.1 Effectiveness of $\mathcal{SI}^*$ as a Dependence Measure

We first illustrate the superiority of $\mathcal{SI}^*$ as a dependence measure by comparing against the following state-of-the-art measures: Mutual Information (MI) (Shannon, 1948; Kraskov et al., 2004), Sliced Mutual Information (SI) (Goldfeld and Greenewald, 2021), distributional sliced Wasserstein Dependence (WD) (Nguyen et al., 2021), Transport Dependency Coefficient (TD) (a.k.a Transport Correlation) (Nies et al., 2021), Mutual Information Dimension (MID) (Sugiyama and Borgwardt, 2013), Maximal Information Coefficient (Reshef et al., 2011), and Distance Correlation (dCor) (Székely et al., 2007). We perform ROC curve analysis to examine precision and recall in recognizing complex relationship types with the existence of uniformly distributed noise. We test on synthetic data that comprises positive samples that depict extremely complicated structures of $P_{X,Y} \in \mathcal{P}(\mathbb{R}^3 \times \mathbb{R}^3)$. In this setting, we conduct 10 statistical tests and cover a wide range of structures, including sinusoidal features, elliptical surfaces, spherical harmonic surfaces, and hyper-hypocycloids. We provide the full details (e.g. the mappings formulae) in Appendix D.2.

Results are reported in Figure 2 as a function of the noise ratio. Notably, $\mathcal{SI}^*$ surpassed all other dependency measures in discerning structure from noise and sometimes by a significant margin (up to 30% improvement in accuracy). Although $\mathcal{SI}^*$ has shown similar performance to some of the baselines in the linear and parabolic settings, a clear superiority of $\mathcal{SI}^*$ is manifested in the eight complicated geometries, such as the spherical harmonics and sinusoidals. As a result, Figure 2 clearly shows the advantages of our slicing optimality in highly non-trivial settings.

## 6.2 Sample and Slicing Efficiency of $\mathcal{SI}^*$

Supporting our theory in Section 3.3, we show that $\mathcal{SI}^*$ scales efficiently to higher dimensions. We test against SI and MI estimators (Kraskov et al., 2004) on two structures of common signal and elliptical. We calculate the AUC of the ROC while varying the number of samples or projections. Figure 3 shows that $\mathcal{SI}^*$ attains the perfect accuracy with less number of slices compared to the uniform-sampling-based baselines. As such, this empirical result supports our claim that $\mathcal{SI}^*$ succeeds in detecting the dependence between the variables with an improved sample and projection efficiency. Appendix B.2 provides additional experiments in higher dimensions.

## 6.3 Result on Representation Learning

Representation learning employs MI to discover useful representations by training a neural network encoder to maximize MI between its inputs and outputs. For example, Hjelm et al. (2019) introduces Deep InfoMax (DIM), a popular method for learning unsupervised representations, which improves the learning by including knowledge about the local structure of the input into the objective. We follow DIM settings and substitute MI with $\mathcal{SI}$ and $\mathcal{SI}^*$ as the information measure, respectively. We test these three methods along with BiGAN (Donahue et al., 2016) on the STL-10 (Coates et al., 2011) and CIFAR10 (Krizhevsky et al., 2009) datasets, which consist of high-dimensional images. Results in Tables 1 and 2 show that $\mathcal{SI}^*$ outperforms the baselines for all classifiers: high-level vector

representation ($Y$), the output of the previous fully-connected layer (fc) and the last convolutional layer (conv). As such, this experiment highlights the scalability aspect of $\mathcal{SI}^*$ with high-dimensional variables not to mention its excellent performance on a challenging ML task.

Table 1: Classification accuracy (%) on STL-10

|  | conv | fc | Y |
|---|---|---|---|
| BiGAN | 71.53 | 67.18 | 58.48 |
| DIM (MI) | 69.15 | 63.81 | 61.92 |
| DIM (SI) | 74.54 | 71.34 | 68.90 |
| DIM (SI*) | **76.89** | **71.67** | **70.04** |

Table 2: Classification accuracy (%) on CIFAR10

|  | conv | fc | Y |
|---|---|---|---|
| BiGAN | 62.57 | 62.74 | 52.54 |
| DIM (MI) | 72.66 | **70.66** | 64.71 |
| DIM (SI) | 74.37 | 70.23 | 65.99 |
| DIM (SI*) | **77.01** | 70.39 | **69.04** |

## 6.4 Result on Reinforcement Learning

Previous frameworks have studied MI objectives as regularizers to the reinforcement learning (RL) objective that involves high-dimensional variables of states, actions, and rewards (Nachum et al., 2019; Schwarzer et al., 2021). Since it is practically difficult to estimate MI in high dimensions, most works resort to learning representations of the high-dimensional data by projecting them into low-dimensional embeddings. Thanks to the slicing technique, these representations are no longer needed when using $\mathcal{SI}^*$. We empirically evaluate the scalability of $\mathcal{SI}^*$ with an RL task, where we adapt the forward information objective in Rakelly et al. (2021) as:

$$\mathbb{J}^\pi_{\mathcal{SI}^*} = \max_\pi \mathcal{SI}^*(S_{t+1}; [A_t, S_t]), \tag{7}$$

where $\pi$ denotes the agent's policy, $A_t \sim \pi(S_t)$, $S_{t+1} \sim \mathbb{T}(S_t, A_t)$, and $\mathbb{T}$ is the transition dynamics. For SI, we use the same objective (by replacing $\mathcal{SI}^*$ with the dependence measure $\mathcal{SI}(.;.)$). We run tests on three challenging Gym environments (Brockman et al., 2016), where the environment dimensions (observation dim, action dim) are as follows: Humanoid-v3: $(378, 17)$; Ant-v3: $(113, 8)$; Hopper-v3: $(12, 3)$. Further details can be found in Appendix D.3. We report the results comparing $\mathcal{SI}^*$, SI, and the original paper objective (MI). Figure 4 shows clear gains (up to $25\%$) with a noticeable learning speed of $\mathcal{SI}^*$ compared to the baselines.

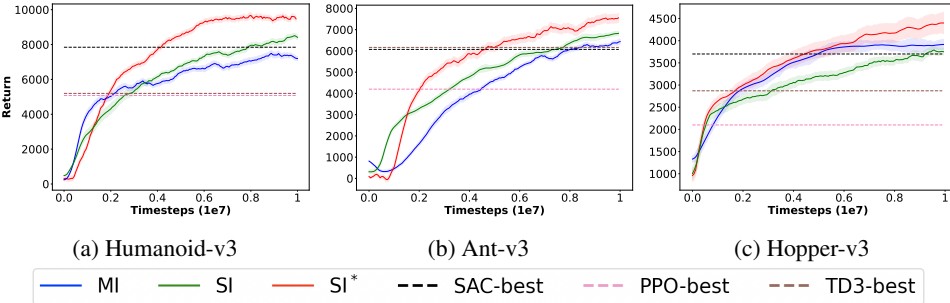

    (a) Humanoid-v3          (b) Ant-v3          (c) Hopper-v3

    —— MI     —— SI     —— SI*     - - - SAC-best     - - - PPO-best     - - - TD3-best

Figure 4: Results on high-dimensional Reinforcement Learning environments show a cutting-edge performance of $\mathcal{SI}^*$ on complex high-dimensional control problems. The mean and variance computed for 12 seeds are shown in the figures.

## 6.5 Behavior Analysis

**Convergence of $\mathcal{SI}^*$.** We validate the convergence rate of $\mathcal{SI}^*$ (Theorem 4). In Figure 5, we show the RMSE between the ground truth $\mathcal{SI}^*$ and the estimated in the setting where $X$ and $Y$ are normal random variables with 5 overlapping entries: $Z \sim \mathcal{N}(0, I_{15})$, $\Lambda \sim \mathcal{N}(0, 0.1I_{10})$, $X = Z_{[1,10]}$, $Y = Z_{[6,15]} + \Lambda$. The heat map in Figure 5a shows convergence when $n, m$ (# samples and # slices, respectively) vary independently, while Figure 5b shows when one parameter is varying and the other is set to $10^3$. The MI estimator used is the Kozachenko–Leonenko estimator (Kraskov et al., 2004). We also test with $X, Y \in \mathbb{R}^d$ with linear dependence to study the effect of the dimension on the convergence. Results are reported in Figure 5c with $n$ and $m$ varying together.

**Computational complexity.** Results in Figure 6 (left) show that the computational complexity of $\mathcal{SI}^*$ is of an order similar to that of $\mathcal{SI}$. Note that, while $\mathcal{SI}^*$ is slower than $\mathcal{SI}$, the difference is

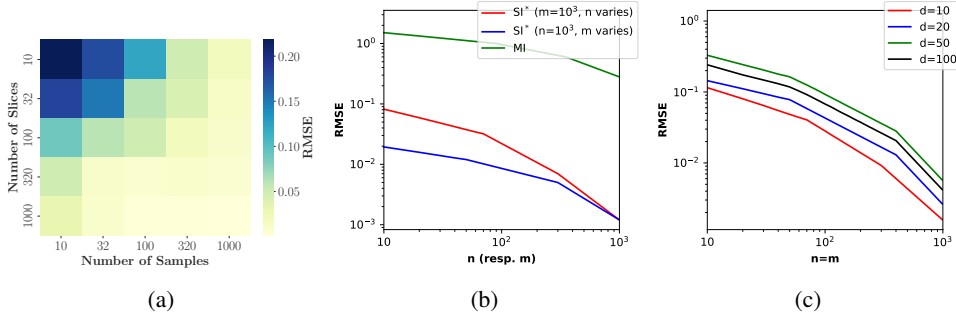

|  (a) | (b) | (c) |
|---|---|---|

Figure 5: Figure shows the RMSE with varying $n$, $m$ and variables dimensions.

insignificant not to mention the clear advantage of $\mathcal{SI}^*$ over other baselines given its superiority in detecting complex dependencies. We also analyze the effect of number of gradient update steps of $f_1$, $f_2$ on the computation speed (Figure 6 (right)). Further results can be found in Appendix B.

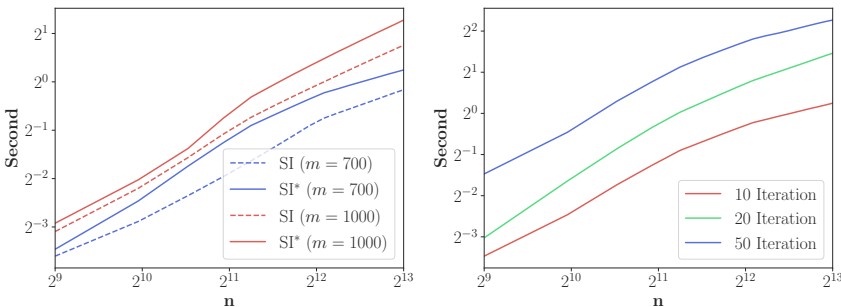

Figure 6: (Left) Computation speed of measures based on the number of samples and slices (dependency is linear and noise ratio $= 0.6$). (Right) Measures with respect to the number of update steps ($m = 700$).

## 7 Concluding Remarks

We presented a novel dependence measure, called $\mathcal{SI}^*$, that is scalable to high dimensions while being efficient regarding time, sample, and slicing complexity. We discussed its theoretical properties, proved that its estimation error depends on the problem dimensions only up to a constant factor, and empirically validated the competence of $\mathcal{SI}^*$ in detecting complicated dependencies against state-of-the-art dependence measures. We further placed $\mathcal{SI}^*$ into modern ML, where we proved the adequacy of our dependence measure on a more onerous set of tasks.

**Limitations.** Since the aim of this paper was to introduce optimality into slicing methods, $\mathcal{SI}^*$ is formalized to use two slicing variables. Although the current setting has proved effective and rather sufficient in complex machine learning scenarios, the extension into multivariate settings is encouraged for future works.

**Broader impact.** The findings and methodologies developed through this work can significantly improve the way we understand and analyze systems with complex relations. We believe this work will have noticeable impact on broader communities, since the study of quantifying information is active in research areas including physics, statistics, computational biology, economics, and neuroscience. At its current form, this work is mainly theoretical and does not impose any negative societal impact.

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
