# On Slicing Optimality for Mutual Information

**Ammar Fayad**[*]
MIT

**Majd Ibrahim**
HIAST

## A  Proofs & Mathematical Details

### A.1  Proof of Theorem 1

*Proof.* We first prove that the set of slicing policies $\Sigma$ is tight in $\mathcal{P}(\mathbb{S}^{d_x-1} \times \mathbb{S}^{d_y-1})$.

**Lemma 1** ((Villani, 2009)). *Let $\mathcal{X}$ and $\mathcal{Y}$ be Polish spaces. Let $P \subset \mathcal{P}(\mathcal{X})$ and $Q \subset \mathcal{P}(\mathcal{Y})$ be tight subsets of the respective probability spaces. The set $\Sigma(P \times Q)$ of policies whose marginals lie in $P$ and $Q$, respectively, is tight in $\mathcal{P}(\mathcal{X} \times \mathcal{Y})$.*

*Proof.* Let $\mu \in P, \nu \in Q$, and $\sigma \in \Sigma(\mu, \nu)$. We assume that, for any $\epsilon > 0$, there exists compact sets $K_\epsilon \subset \mathcal{X}, L_\epsilon \subset \mathcal{Y}$, independent of the choice of $\mu$ and $\nu$, such that $\mu[\mathcal{X} \setminus K_\epsilon] \leq \epsilon$ and $\nu[\mathcal{Y} \setminus L_\epsilon] \leq \epsilon$. Note that for any coupling $(u, v)$ of $(\mu, \nu)$:

$$\Pr[(u,v) \notin K_\epsilon \times L_\epsilon] \leq \Pr[u \notin K_\epsilon] + \Pr[v \notin L_\epsilon] \leq 2\epsilon.$$

The conclusion follows given that $\epsilon$ is arbitrary, and that $K_\epsilon \times L_\epsilon$ is compact in $\mathcal{X} \times \mathcal{Y}$. $\square$

Since $\mathbb{S}^{d-1}$ is compact, the families of measures $\Sigma_{\omega_X} \subseteq \mathcal{P}(\mathbb{S}^{d_x-1})$ and $\Sigma_{\omega_Y} \subseteq \mathcal{P}(\mathbb{S}^{d_y-1})$ are tight. Therefore, $\Sigma$ is tight in $\mathcal{P}(\mathbb{S}^{d_x-1} \times \mathbb{S}^{d_y-1})$ as dictated by Lemma 1. Furthermore, by Prokhorov's theorem (Billingsley, 2013)[2], the closure of this set is sequentially compact in the topology of weak convergence.[3]

**Lemma 2.** *For random variables $S, Q$ satisfying our assumptions, $\mathcal{I}(S; Q)$ is continuous in the topology of weak convergence.*

*Proof.* Findings similar to this lemma have been studied in (Piera and Parada, 2009; Godavarti and Hero, 2004; Ghourchian et al., 2017), we present the outline of our argument into three steps:

**Step 1:** For any r.v. $X$ obeying our former assumptions, $H(X)$ is bounded: Since $\sup_x f(x) < \infty, H(X) = -\int f \log f \geq -\log \sup_x f$,. To show that $H(X)$ is bounded from above, we use the fact from Godavarti and Hero (2004), that is for any $\epsilon > 0, |\int_{\|x\|>K} f(x) \log f(x) dx| < \epsilon$ for sufficiently large $K$. It follows that $H(X) \leq |\int_{\|x\| \leq K} f(x) \log f(x) dx| + |\int_{\|x\|>K} f(x) \log f(x) dx| \leq (\sup_x f(x) \log f(x)) \mathrm{vol}(\|x\| \leq K) + \epsilon < \infty$.

**Step 2:** For any sequence $(\theta_n^\top X)_{n \in \mathbb{N}}$ where $\theta_n^\top X \to \theta^\top X$ and the pdfs $f_n := p_{\theta_n^\top X} \to f := p_{\theta^\top X}$,

$$\limsup_{n \to \infty} H(\theta_n^\top X) \leq H(\theta^\top X).$$

This step follows from the observation that, $\forall \epsilon > 0$, there exists sufficiently large $M, N$ s.t. if $\mathcal{A}_M = \mathbb{R} \setminus \bigcup_{k=M}^{\infty}\{x : f_k(x) > \Omega\}$ for sufficiently large $\Omega > \sup_x f(x)$, then for all $n > N$,

---

[*]afayad@mit.edu

[2]Prokhorov's theorem (Billingsley, 2013) states that if $(\mathcal{X}, \rho)$ is a separable metric space, then $K \subset \mathcal{P}(\mathcal{X})$ is tight iff the closure of $K$ is sequentially compact in $\mathcal{P}(\mathcal{X})$ with respect to the weak convergence.

[3]A sequence of probability measures $\pi_n \in \mathcal{P}(\mathcal{X})$ converges weakly to $\pi$, and we write $\pi_n \to \pi$, if for any $f \in C_b^0(\mathcal{X})$ we have $\int_{\mathcal{X}} f d\pi_n \to \int_{\mathcal{X}} f d\pi$, where $C_b^0(\mathcal{X})$ is the space of bounded and continuous functions on $\mathcal{X}$.

37th Conference on Neural Information Processing Systems (NeurIPS 2023).

$|\int_{\mathcal{A}_M} f_n \log f_n - \int_{\mathcal{A}_M} f \log f| < \epsilon$. Also, we have $H(\theta_n^\top X) + \int_{\mathcal{A}_M} f_n \log f_n \leq H(\theta^\top X) + \int_{\mathcal{A}_M} f \log f$.

**Step 3:** For any sequence $(\theta_n^\top X)_{n \in \mathbb{N}}$ where $\theta_n^\top X \to \theta^\top X$ and the pdfs $f_n := p_{\theta_n^\top X} \to f := p_{\theta^\top X}$,

$$\liminf_{n \to \infty} H(\theta_n^\top X) \geq H(\theta^\top X).$$

We also arrive at the continuity of the differential joint entropy by considering the joint pdf and proceeding in the same manner as before. Lastly, since $\mathcal{I}(S; Q) = H(S) + H(Q) - H(S, Q)$, the conclusion follows. $\qquad\square$

Next, let $(\sigma_k)_{k \in \mathbb{N}}$ be a sequence of probability measures in $\mathcal{P}(\mathbb{S}^{d_x-1} \times \mathbb{S}^{d_y-1})$ such that $\oint \mathcal{I}(\theta^\top X; \varphi^\top Y) d\sigma_k$ converges to the supremum in Eq. (3); since $\Sigma$ is compact, we know that $\sigma_k$ converges to some $\sigma$. Then, because of the continuity (Lemma 2), we can express $\mathcal{I}$ as the pointwise limit of a nonincreasing collection $(f_l)_{l \in \mathbb{N}}$ of continuous real-valued functions, and

$$\oint \mathcal{I}(\theta^\top X; \varphi^\top Y) d\sigma = \lim_{l \to \infty} \oint f_l(\theta, \phi) d\sigma = \lim_{l \to \infty} \lim_{k \to \infty} \oint f_l d\sigma_k \geq \limsup_{k \to \infty} \oint \mathcal{I}(\theta^\top X; \varphi^\top Y) d\sigma_k.$$

Thus, $\sigma$ is maximizing. $\qquad\blacksquare$

**Remark 1.** *We could proceed differently by imposing stronger assumptions using the following lemma:*

**Lemma 3** ((Yang and Chen, 2018))**.** *For a continuous $P_0 \in \mathcal{P}(\mathbb{R})$ and $\delta_0 > 0$, let $\mathbb{B}(P_0, \delta_0) = \{P \in \mathcal{P}(\mathbb{R}) : d_{\mathcal{L}}(P, P_0) \leq \delta_0\}$.[4] The robust KL divergence defined as $\inf_{\nu \in \mathbb{B}(P_0, \delta_0)} KL(\mu || \nu)$ is continuous in $\mu$ with respect to the weak convergence.*

*Proof. We briefly discuss the outline of the proof for the sake of completeness. The interested reader is encouraged to see Yang and Chen (2018) for full proof.*

***Step 1: Lower semicontinuity.*** *Define $\overline{\mathbb{B}}(P_0, \delta_0) = \{P \in \mathcal{M} : d_{\mathcal{L}}(P, P_0) \leq \delta_0\}$, where $\mathcal{M}$ is the space of finitely additive and nonnegative set functions on $\mathbb{R}$ and its Borel $\sigma$-algebra. Note that $\overline{\mathbb{B}}(P_0, \delta_0)$ is closed with respect to the weak convergence, thus is compact since $\mathcal{M}$ is compact (Loeve, 2017). It also can be shown that $\inf_{\nu \in \mathbb{B}(P_0, \delta_0)} KL(\mu || \nu) = \inf_{\nu \in \overline{\mathbb{B}}(P_0, \delta_0)} KL(\mu || \nu)$. Now assume that $\mu_n \to \mu$. We know from (Yang and Chen, 2018) that $\exists \nu_n \in \mathbb{B}(P_0, \delta_0), KL(\mu_n || \nu_n) = \inf_{\nu \in \mathbb{B}(P_0, \delta_0)} KL(\mu_n || \nu)$. On the other hand, since $\overline{\mathbb{B}}$ is compact, there exists a subsequence of $\nu_n$ that converges to $\nu_0 \in \overline{\mathbb{B}}$. Thus, $KL(\mu || \nu_0) \leq \liminf_{n \to \infty} KL(\mu_n || \nu_n)$ because of the lower semicontinuity of $KL(\cdot || \cdot)$ and $(\mu_n, \nu_n) \to (\mu, \nu_0)$. In conclusion,*

$$\inf_{\nu \in \mathbb{B}(P_0, \delta_0)} KL(\mu || \nu) = \inf_{\nu \in \overline{\mathbb{B}}(P_0, \delta_0)} KL(\mu || \nu) \leq KL(\mu || \nu_0)$$

$$\leq \liminf_{n \to \infty} KL(\mu_n || \nu_n) = \liminf_{n \to \infty} \inf_{\nu \in \mathbb{B}(P_0, \delta_0)} KL(\mu_n || \nu).$$

***Step 2: Upper semicontinuity.*** *The argument here depends on two important facts:*

1. *For $\delta, \delta_0 > 0$ and $\mu_0, P_0 \in \mathcal{P}(\mathbb{R})$, let $\mu_x^\delta(t) = \mathbf{1}_{x < t} \max(0, \mu_0(t - \delta) - \delta) + \mathbf{1}_{x \geq t} \min(1, \mu_0(t + \delta) + \delta)$. We have:*

$$\sup_{\mu \in \mathbb{B}(\mu_0, \delta_0)} \inf_{\nu \in \mathbb{B}(P_0, \delta_0)} KL(\mu || \nu) = \sup_{x \in \mathbb{R}} \inf_{\nu \in \mathbb{B}(P_0, \delta_0)} KL(\mu_x^\delta || \nu).$$

2. *The robust KL divergence is continuous in $\delta_0 > 0$, and its supremum is bounded:*

$$\sup_{P_0, \mu \in \mathcal{P}(\mathbb{R})} \inf_{\nu \in \mathbb{B}(P_0, \delta_0)} KL(\mu || \nu) = \log \frac{1}{\delta_0}.$$

---

[4]$d_{\mathcal{L}}$ is the Lévy metric (Levy, 1955) between probability measures on the real line and is defined for $F, G : \mathbb{R} \to [0, 1]$ as $d_{\mathcal{L}}(F, G) = \inf\{\epsilon > 0 : F(x - \epsilon) - \epsilon \leq G(x) \leq F(x + \epsilon) + \epsilon, \forall x \in \mathbb{R}\}$.

*Based on the above, we can show that*

$$\limsup_{\delta \to 0} \inf_{x \in \mathbb{R}} \inf_{\nu \in \mathbb{B}(P_0, \delta_0)} KL\big(\mu_x^\delta || \nu\big) \leq \lim_{\delta_1 \to 0} \left( \log \frac{1}{1 - \delta_1} + \inf_{\nu \in \mathbb{B}(P_0, \delta_0 - \delta_1)} KL\big(\mu_0 || \nu\big) \right)$$
$$= \inf_{\nu \in \mathbb{B}(P_0, \delta_0)} KL\big(\mu_0 || \nu\big). \qquad \square$$

*The existence of an optimal slicing policy follows then by assuming that* $\mathcal{I}(\theta^\top X; \varphi^\top Y) = \inf_{Q_{X,Y} \in \mathbb{B}(\theta_\#^* P_X \otimes \varphi_\#^* P_Y, \delta_0)} KL\big((\theta^* \times \varphi^*)_\# P_{X,Y} || Q_{X,Y}\big) \geq 0$ *for small enough* $\delta_0$ *(continuity at 0).*

## A.2 Proof of Theorem 2

1. $\mathcal{SI}^*(X; Y) = \mathcal{SI}^*(Y; X) \geq 0$ is straightforward by the properties of the mutual information. $\square$

2. Since the set of slicing policies $\Sigma$ is compact and tight in $\mathcal{P}(\mathbb{S}^{d_x - 1} \times \mathbb{S}^{d_y - 1})$ (see proof in Appendix A.1), we may assume that $\sigma^* \in \Sigma$ is a slicing policy that maximizes the information functional $\oint \mathcal{I}(\theta^\top X; \phi^\top Y) d\sigma$ for a given coupling $P_{X,Y} \in \mathcal{P}(\mathbb{R}^{d_x} \times \mathbb{R}^{d_y})$. Now, if $X$ and $Y$ are independent then $\mathcal{I}(X; Y) = 0$, since $\mathcal{I}(\theta^\top X; \phi^\top Y) \leq \mathcal{I}(X; Y)$ for all $\theta, \phi$ (Theorem 3), then $\mathcal{SI}^*(X; Y) = 0$. Conversely, let $\varphi_{X,Y}(t, s) \overset{\text{def}}{=} \mathbb{E}\big[e^{itX + isY}\big]$ be the joint characteristic function. If

$$\mathcal{SI}^*(X; Y) = \oint_{\mathbb{S}^{d_x - 1} \times \mathbb{S}^{d_y - 1}} \mathcal{I}(X_\theta; Y_\phi) d\sigma^*(\theta, \phi) = 0$$

where $X_\theta \overset{\text{def}}{=} \theta^\top X$ and $Y_\phi \overset{\text{def}}{=} \phi^\top Y$, it follows that $(X_\theta, Y_\phi)$ are independent for all $(\theta, \phi) \in \mathbb{S}^{d_x - 1} \times \mathbb{S}^{d_y - 1}$, or, equivalently, $\varphi_{X_\theta, Y_\phi}(t, s) = \varphi_{X_\theta}(t) \varphi_{Y_\phi}(s)$ for all $t, s \in \mathbb{R}$. The latter is the same as

$$\varphi_{X,Y}(t\theta, s\phi) = \varphi_X(t\theta) \varphi_Y(s\phi), \quad \forall t, s \in \mathbb{R}, \ \theta \in \mathbb{S}^{d_x - 1}, \ \phi \in \mathbb{S}^{d_y - 1}.$$

If we change variables $t' = t\theta$ and $s' = s\phi$, we arrive at: $\varphi_{X,Y}(t', s') = \varphi_X(t') \varphi_Y(s'), \quad \forall t' \in \mathbb{R}^{d_x}, \ s' \in \mathbb{R}^{d_y}$, meaning that $X$ and $Y$ are independent. $\square$

3. This follows from the continuity result established in Lemma 2. $\square$

4. The mutual information $\mathcal{I}(X; Y)$ can be expressed as $\mathcal{I}(X; Y) = \text{KL}\big(P_{X,Y} || P_X \otimes P_Y\big)$,

$$\mathcal{SI}^*(X; Y) = \sup_\sigma \mathbb{E}_{(\Theta, \Phi) \sim \sigma} \left[ \text{KL}\big((\Theta^* \times \Phi^*)_\# P_{X,Y} || \Theta_\#^* P_X \otimes \Phi_\#^* P_Y\big) \right]$$
$$= \sup_\sigma \text{KL}\big(\sigma \otimes (\Theta^* \times \Phi^*)_\# P_{X,Y} || \sigma \otimes \Theta_\#^* P_X \otimes \Phi_\#^* P_Y\big).$$

The expression follows from the fact that the joint distribution of $(\Theta^\top X, \Phi^\top Y)$ is $(\Theta^* \times \Phi^*)_\# P_{X,Y}$, while the corresponding conditional marginals are $\Theta_\#^* P_X$ and $\Phi_\#^* P_Y$, respectively. $\square$

### A.2.1 Proof of Corollary 1

*Proof.* This result follows directly from the characterization of the KL divergence using a discriminator function (Mescheder et al., 2017; Sønderby et al., 2016; Ghimire et al., 2021). $\blacksquare$

### A.2.2 Proof of Corollary 2

*Proof.* We first define the sets $\mathcal{F}_{\omega_X} \overset{\text{def}}{=} \{f : f \in \mathcal{C}(\mathbb{S}^{d_x - 1} \times \mathbb{S}^{d_y - 1}, \mathbb{S}^{d_x - 1}), \ f_\# \gamma(\mathbb{S}^{d_x - 1} \times \mathbb{S}^{d_y - 1}) \in \Sigma_{\omega_X}\}$ and $\mathcal{F}_{\omega_Y} \overset{\text{def}}{=} \{f : f \in \mathcal{C}(\mathbb{S}^{d_x - 1} \times \mathbb{S}^{d_y - 1}, \mathbb{S}^{d_y - 1}), \ f_\# \gamma(\mathbb{S}^{d_x - 1} \times \mathbb{S}^{d_y - 1}) \in \Sigma_{\omega_Y}\}$.

Note that for any optimal slicing policy $\sigma$, we can find a Borel map $f : \mathbb{S}^{d_x - 1} \times \mathbb{S}^{d_y - 1} \to \mathbb{S}^{d_x - 1} \times \mathbb{S}^{d_y - 1}$ such that $\sigma = f_\# \gamma$. Based on this observation and after defining $\mathcal{I}(\theta, \varphi) \overset{\text{def}}{=} \mathcal{I}(\theta^\top X, \varphi^\top Y)$, we obtain

$$\mathcal{SI}^*(X; Y) = \oint \mathcal{I}(\theta, \varphi) d\sigma(\theta, \varphi) = \oint \mathcal{I} \circ f(\psi, \upsilon) d\gamma(\psi, \upsilon).$$

Now since the marginals of $\sigma$ lie in $\Sigma_X$ and $\Sigma_Y$, we can write $f$ as $(f_1, f_2)$ where $f_1 \in \mathcal{F}_{\omega_X}$ and $f_2 \in \mathcal{F}_{\omega_Y}$. Finally,

$$
\begin{aligned}
\mathcal{SI}^*(X;Y) &= \sup_{f_1 \in \mathcal{F}_{\omega_X}, f_2 \in \mathcal{F}_{\omega_Y}} \oint \mathcal{I}(f_1(\psi, \upsilon)^\top X; f_2(\psi, \upsilon)^\top Y) d\boldsymbol{\gamma}(\psi, \upsilon) \\
&= \sup_{f_1 \in \mathcal{F}_{\omega_X}, f_2 \in \mathcal{F}_{\omega_Y}} \mathbb{E}_{(\psi,\upsilon) \sim \boldsymbol{\gamma}(\mathbb{S}^{d_x-1} \times \mathbb{S}^{d_y-1})} \left[ \mathcal{I}(f_1(\psi, \upsilon)^\top X; f_2(\psi, \upsilon)^\top Y) \right] \\
&= \sup_{f_1 \in \mathcal{F}_{\omega_X}, f_2 \in \mathcal{F}_{\omega_Y}} \mathrm{KL}\left( \boldsymbol{\gamma} \otimes (f_1 \times f_2)(\Psi, \Upsilon)^*_\# P_{X,Y} \| \boldsymbol{\gamma} \otimes f_1(\Psi, \Upsilon)^*_\# P_X \otimes f_2(\Psi, \Upsilon)^*_\# P_Y \right).
\end{aligned}
$$

Using the Donsker-Varadhan representation (Donsker and Varadhan, 1975) of the KL divergence, we arrive at our conclusion:

$$
\begin{aligned}
\mathcal{SI}^*(X;Y) = \sup_{T, f_1, f_2} \Big\{ &\mathbb{E}\left[ T(\Psi, \Upsilon, f_1(\Psi, \Upsilon)^\top X, f_2(\Psi, \Upsilon)^\top Y) \right] \\
&- \log \mathbb{E}\left[ \exp\left( T(\Psi, \Upsilon, f_1(\Psi, \Upsilon)^\top \overline{X}, f_2(\Psi, \Upsilon)^\top \overline{Y}) \right) \right] \Big\}.
\end{aligned} \tag{1}
$$

■

### A.3  Proof of Theorem 3

*Proof.* The inequality $\mathcal{I}(X;Y) \geq \mathcal{SI}^*(X;Y)$ follows directly from the Data Processing Inequality (DPI), where we note that:

$$
\mathcal{I}(\theta^\top X; \varphi^\top Y) \leq \mathcal{I}(X;Y) \quad \forall (\theta, \varphi) \in \mathbb{S}^{d_x-1} \times \mathbb{S}^{d_y-1},
$$

and the conclusion follows. For the second inequality, we need to prove that $\mathcal{SI}^*(X;Y) \geq \oint \mathcal{I}(\theta^\top X; \varphi^\top Y) d\boldsymbol{\gamma}(\theta) \otimes \boldsymbol{\gamma}(\varphi) \stackrel{\text{def}}{=} \mathcal{SI}(X;Y)$. Thus, it suffices to show that $\boldsymbol{\gamma}(\mathbb{S}^{d_x-1}) \in \Sigma_{\omega_X}$ and $\boldsymbol{\gamma}(\mathbb{S}^{d_y-1}) \in \Sigma_{\omega_Y}$:

$$
\mathbb{E}_{x,x' \sim \boldsymbol{\gamma}}[\arccos|x^\top x'|] = \oint_{(\mathbb{S}^{d-1})^2} \arccos|x^\top x'| d\boldsymbol{\gamma}(x) \otimes \boldsymbol{\gamma}(x') = \oint_{\mathbb{S}^{d-1}} \arccos|x^\top \alpha| d\boldsymbol{\gamma}(x),
$$

where $\alpha \in \mathbb{S}^{d-1}$. The last equation follows from the fact that since $\boldsymbol{\gamma}$ is uniform over $\mathbb{S}^{d-1}$, the integral $\oint \arccos|x^\top x'| d\boldsymbol{\gamma}(x)$ is the same for all $x'$.

Set $\alpha = (0, 0, \ldots, 0, 1)$ and denote by $x_i$ the $i$-th entry of $x$:

$$
I \stackrel{\text{def}}{=} \oint_{\mathbb{S}^{d-1}} \arccos|x_d| d\boldsymbol{\gamma}(x) = \frac{1}{A} \oint \arccos|x_d| dA,
$$

where $dA$ is the surface area element of $\mathbb{S}^{d-1}$ and $A = \int_{\mathbb{S}^{d-1}} dA$. We now represent $x$ in terms of angular parameters $\phi_1 \in [0, 2\pi); \phi_2, \ldots, \phi_{d-1} \in [0, \pi)$ as following:

$$
\begin{aligned}
x_d &= \cos(\phi_{d-1}) \\
x_{d-1} &= \sin(\phi_{d-1}) \cos(\phi_{d-2}) \\
x_{d-2} &= \sin(\phi_{d-1}) \sin(\phi_{d-2}) \cos(\phi_{d-3}) \\
&\vdots \\
x_2 &= \sin(\phi_{d-1}) \sin(\phi_{d-2}) \ldots \sin(\phi_2) \cos(\phi_1) \\
x_1 &= \sin(\phi_{d-1}) \sin(\phi_{d-2}) \ldots \sin(\phi_2) \sin(\phi_1)
\end{aligned}
$$

This allows a more feasible definition of surface area element: $dA = \sqrt{\det G} d\phi_1 \ldots d\phi_{d-1}$, where $G$ is the metric tensor; for $1 \leq i, j \leq d-1$:

$$
G_{ij} = \sum_{k=1}^{d} \frac{\partial x_k}{\partial \phi_i} \frac{\partial x_k}{\partial \phi_j}
$$

Consequently, $dA = \sin^{d-2}(\phi_{d-1})\sin^{d-3}(\phi_{d-2})\ldots\sin(\phi_2)d\phi_1\ldots d\phi_{d-1}$. Back to the integral, and after applying Fubini's theorem,

$$I = \frac{1}{A}\left(\int_0^\pi \arccos|\cos\phi_{d-1}|\sin^{d-2}(\phi_{d-1})d\phi_{d-1}\right)\left(\int_0^\pi \sin^{d-3}(\phi_{d-2})d\phi_{d-2}\right)\ldots\left(\int_0^{2\pi} d\phi_1\right)$$

On the other hand,

$$A = \left(\int_0^\pi \sin^{d-2}(\phi_{d-1})d\phi_{d-1}\right)\left(\int_0^\pi \sin^{d-3}(\phi_{d-2})d\phi_{d-2}\right)\ldots\left(\int_0^{2\pi} d\phi_1\right)$$

Hence,

$$
\begin{aligned}
I &= \left(\int_0^\pi \arccos|\cos\phi_{d-1}|\sin^{d-2}(\phi_{d-1})d\phi_{d-1}\right)\Big/\left(\int_0^\pi \sin^{d-2}(\phi_{d-1})d\phi_{d-1}\right)\\
&= \left(\int_0^{\frac{\pi}{2}}\phi\big(\sin^{d-2}(\phi)+\cos^{d-2}(\phi)\big)d\phi\right)\Big/\left(\int_0^\pi \sin^{d-2}(\phi_{d-1})d\phi_{d-1}\right)\\
&= \left(\frac{1}{2}\int_0^{\frac{\pi}{2}}\frac{\pi}{2}\big(\sin^{d-2}(\phi)+\cos^{d-2}(\phi)\big)d\phi\right)\Big/\left(\int_0^\pi \sin^{d-2}(\phi_{d-1})d\phi_{d-1}\right)\\
&= \left(\frac{\pi}{4}\left(\int_0^\pi \sin^{d-2}(\phi_{d-1})d\phi_{d-1}\right)\right)\Big/\left(\int_0^\pi \sin^{d-2}(\phi_{d-1})d\phi_{d-1}\right)\\
&= \frac{\pi}{4}.
\end{aligned}
$$

In conclusion, for a given $\omega \geq \pi/4$, the set $\Sigma_\omega$ contains the uniform distribution, which directly implies that $\mathcal{SI}^*(X;Y) \geq \mathcal{SI}(X;Y)$. ∎

### A.4 Proof of Theorem 4

*Proof.* Let $(\Theta, \Phi) \sim \sigma$ and define $\mathcal{I}^{XY}(\theta,\phi) := \mathcal{I}(\theta^\top X; \phi^\top Y)$. From here, we proceed similarly to (Corollary S4 Nadjahi et al., 2020; Goldfeld and Greenewald, 2021, Theorem. 1). By the triangle inequality, we have

$$\left|\mathcal{SI}^*(X;Y) - \widehat{\mathcal{SI}^*}_{n,m}(X;Y)\right| \leq \left|\mathcal{SI}^*(X;Y) - A\right| + \left|A - \widehat{\mathcal{SI}^*}_{n,m}(X;Y)\right|,$$

where

$$A = \sup_\sigma \frac{1}{m}\sum_{i=1}^m \mathcal{I}^{XY}(\Theta_i, \Phi_i) = \sup_{f_1,f_2}\frac{1}{m}\sum_{i=1}^m \mathcal{I}^{XY}(f_1(\Psi_i,\Upsilon_i), f_2(\Psi_i,\Upsilon_i)),$$

$f_1 \in \mathcal{F}_{\omega_X}$, $f_2 \in \mathcal{F}_{\omega_Y}$, and $(\Psi, \Upsilon) \sim \boldsymbol{\gamma}(\mathbb{S}^{d_x-1} \times \mathbb{S}^{d_y-1})$.

For the first term, since $\{(\Theta_i, \Phi_i)\}_{i=1}^m$ are i.i.d. and $\mathbb{E}\big[\mathcal{I}^{XY}(\Theta,\Phi)\big] = \mathcal{SI}^*(X;Y)$, we obtain

$$
\mathbb{E}\left[\left|\mathcal{SI}^*(X;Y) - \frac{1}{m}\sum_{i=1}^m \mathcal{I}^{XY}(\Theta_i,\Phi_i)\right|\right] \leq \sqrt{\mathbb{E}\left[\left(\mathcal{SI}^*(X;Y) - \frac{1}{m}\sum_{i=1}^m \mathcal{I}^{XY}(\Theta_i,\Phi_i)\right)^2\right]}
$$

$$
= \sqrt{\frac{1}{m}\mathrm{Var}\big(\mathcal{I}^{XY}(\Theta,\Phi)\big)} \leq \frac{U}{2\sqrt{m}}.
$$

The first inequality is a direct application of Cauchy-Schwarz inequality while the second follows from Popoviciu's inequality (Popoviciu, 1935). We should mention that $U \propto \sqrt{\|F_{XY}\|_{\mathrm{op}}\max(\|\Sigma_X\|_{\mathrm{op}}, \|\Sigma_Y\|_{\mathrm{op}})}(d_x^{-1} + d_y^{-1})^{1/2}$, where $F_{XY}$ is the Fisher information matrix of $P_{XY}$ and $\|.\|_{\mathrm{op}}$ is the operational norm of a matrix. $\Sigma_X, \Sigma_Y$ are the marginal covariance matrices of $P_{XY}$. This proportionality follows from applying the Efron-Stein inequality on $\mathcal{I}^{XY}(\theta,\phi)$ after showing that it is a Lipschitz function on the Stiefel manifold (unit sphere), see Goldfeld et al. (2022).

For the second term, we write

$$\mathbb{E}\left[\left|\sup_{f_1,f_2} \frac{1}{m}\sum_{i=1}^{m} \mathcal{I}^{XY}(f_1(\Psi_i,\Upsilon_i),f_2(\Psi_i,\Upsilon_i)) - \widehat{\mathcal{SI}^*}_{n,m}(X;Y)\right|\right]$$

$$= \mathbb{E}\left[\left|\sup_{f_1,f_2} \frac{1}{m}\sum_{i=1}^{m} \mathcal{I}^{XY}(f_1(\Psi_i,\Upsilon_i),f_2(\Psi_i,\Upsilon_i)) - \sup_{f_1,f_2}\frac{1}{m}\sum_{i=1}^{m}\widehat{\mathcal{I}}_n^{XY}(f_1(\Psi_i,\Upsilon_i),f_2(\Psi_i,\Upsilon_i))\right|\right]$$

$$\leq \mathbb{E}\left[\sup_{f_1,f_2}\left|\frac{1}{m}\sum_{i=1}^{m}\mathcal{I}^{XY}(f_1(\Psi_i,\Upsilon_i),f_2(\Psi_i,\Upsilon_i)) - \widehat{\mathcal{I}}_n^{XY}(f_1(\Psi_i,\Upsilon_i),f_2(\Psi_i,\Upsilon_i))\right|\right]$$

$$\leq \mathbb{E}\left[\sup_{f_1,f_2}\frac{1}{m}\sum_{i=1}^{m}\left|\mathcal{I}^{XY}(f_1(\Psi_i,\Upsilon_i),f_2(\Psi_i,\Upsilon_i)) - \widehat{\mathcal{I}}_n^{XY}(f_1(\Psi_i,\Upsilon_i),f_2(\Psi_i,\Upsilon_i))\right|\right]$$

$$\leq \mathbb{E}\left[\sup_{f_1,f_2}\max_{\Psi,\Upsilon}\left|\mathcal{I}^{XY}(f_1(\Psi,\Upsilon),f_2(\Psi,\Upsilon)) - \widehat{\mathcal{I}}_n^{XY}(f_1(\Psi,\Upsilon),f_2(\Psi,\Upsilon))\right|\right]$$

$$= \mathbb{E}\left[\left|\mathcal{I}(S,Q) - \widehat{\mathcal{I}}_n(S,Q)\right|\ \Big|\ \mathrm{law}(S,Q) = (f_1^\dagger(\Psi^\ddagger,\Upsilon^\ddagger)^* \times f_2^\dagger(\Psi^\ddagger,\Upsilon^\ddagger)^*)_\# P_{X,Y}\right]$$

$$\leq \sup_{P_{S,Q}}\mathbb{E}\left[\left|\mathcal{I}(S,Q) - \widehat{\mathcal{I}}_n(S,Q)\right|\right] \leq \delta(n).$$

The assumption that the supremum in the last equality is attained at $f_1^\dagger, f_2^\dagger$ follows from the existence of an optimal slicing policy $\sigma^*$ as demonstrated in A.1. ∎

# B  Further Empirical Analysis

## B.1  More on Behavior Analysis

Our results so far indicate that $\mathcal{SI}^*$ is the first scalable dependence measure that can work excellently in complicated scenarios, we now analyze its behavior in what follows:

**Visualization of slicing directions.**
We study the case where: $Z \sim \mathcal{N}(0,I_4), \Lambda \sim \mathcal{N}(0,0.1I_3), X = [Z_1,Z_2,Z_3]^\top$, and $Y = [\sin(Z_2 + Z_3),Z_2,Z_4]^\top + \Lambda$. Figure 1 contains a 3D scatter of samples drawn from the slicing distributions of $\Theta$ (red) and $\Phi$ (green). In (b), the general pattern of $\theta$ is the great circle $\theta_1 = 0$, which is what we would expect since $X_1$ provides no information about $Y$. $\phi$ also shows a pattern of a great circle $\phi_3 = 0$, but with a cluster of slices around $|\phi_1| = 1$ since $Y_1$ possesses significant amount of information on $X$.

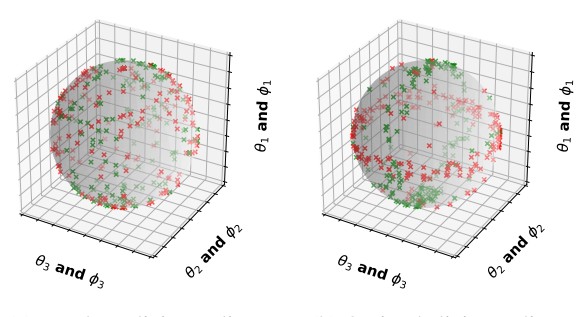

(a) Random slicing policy    (b) Optimal slicing policy

Figure 1: Visualization of slicing policies on unit spheres. Each point represents a slice (red for $\theta$, green for $\phi$).

Table 1: Effect of $\omega_X, \omega_Y$.

| $\omega_Y|\omega_X$ | 0 | $\pi/6$ | $\pi/4$ | $\pi/2$ |
|---|---|---|---|---|
| 0 | 0.71 | 0.80 | 0.91 | 0.93 |
| $\pi/6$ | 0.78 | 0.95 | 0.96 | 0.98 |
| $\pi/4$ | 0.89 | 0.98 | 1.00 | 1.00 |
| $\pi/2$ | 0.92 | 0.96 | 1.00 | 0.96 |

**Illustration of roles of $\omega_X$ and $\omega_Y$.** We conduct experiments to show how $\omega_X$ and $\omega_Y$ (the diversity of slices) affect the accuracy of $\mathcal{SI}^*$. Table 1 reports the AUC of the independence test

of two r.v $X, Y \in \mathbb{R}^{10}$ with linear dependence, where the number of samples and slices used to compute the $\mathcal{SI}^*$ was set to $10^3$ and the noise ratio to 0.6. Generally, the performance improves as $(\omega_X, \omega_Y)$ approaches $(\pi/4, \pi/4)$, which indicates that $(\pi/4, \pi/4)$ balances between slices diversity and concentration around informative regions.

## B.2 Additional Experiments on Sample and Slicing Efficiency

We showcase the scalability of our measure with higher dimensions of input in Figure 2.

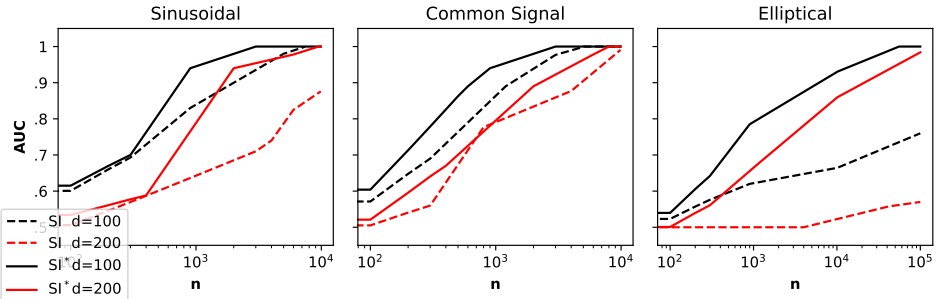

Figure 2: Dependence detection power of $\mathcal{SI}^*$ in higher dimensions. One notices the consistent performance of $\mathcal{SI}^*$, particularly in the more complex geometries cases.

## B.3 More on computational complexity

Results in Table 2 show the computational complexity of the neural estimator $\mathcal{SI}^*_{\mathbb{W}}$.

Table 2: Computation speed of the neural estimators measured by second per minibatch.

| CIFAR10: | $n = 100$ | $n = 300$ | Hopper-v3: | $n = 128$ | $n = 256$ |
|---|---|---|---|---|---|
| MI | 0.39 | 0.75 | MI | 0.25 | 0.81 |
| $\mathcal{SI}$ | 0.42 | 0.98 | $\mathcal{SI}$ | 0.28 | 0.89 |
| $\mathcal{SI}^*$ | 0.47 | 1.15 | $\mathcal{SI}^*$ | 0.30 | 1.03 |

## C  Pseudocode for Estimators

For the neural estimator $\mathcal{SI}^*_{\mathbb{W}}$ used in our representation and reinforcement learning experiments, we proceed as follows: as in Equation (7), use the minibatch $\mathcal{D}$ to calculate $\nabla_{\mathbb{W}} F(\mathbb{W})$, where $F(\mathbb{W}) = \frac{1}{n}\sum_{k=1}^{n} T(.) - \log \frac{1}{n}\sum_{k=1}^{n} \exp(T(.)) + \lambda_1 \left(\frac{1}{n^2}\sum_{k,j} \arccos|.| - \omega_X\right) + \lambda_2 \left(\frac{1}{n^2}\sum_{k,j} \arccos|.| - \omega_Y\right)$. After performing a number of stochastic gradient ascent steps, we arrive at the fixed point in the parameter space, which we denote it by $\mathbb{W}^*$. The estimate of $\mathcal{SI}^*_{\mathbb{W}}$ is then $F(\mathbb{W}^*)$. Next, we present the pseudocode of $\widehat{\mathcal{SI}^*}_{n,m}$ which we studied in Appendix H.3, and we shall stick to the same choice of notations..

## D  Implementation Details

All experiments were performed on a high performance computing system with a SLURM job scheduler (Yoo et al., 2003). The compute nodes used each has two NVIDIA Volta V100 GPUs, and a dual socket Intel Xeon Gold 6248 processor with 20 cores each socket.

### D.1 Details of the Motivating Example

1. To satisfy the first criterion:
   - $\theta_1, \phi_1 \sim \mathcal{N}(0, \epsilon)$ with probability $p >> 0.5$ and sufficiently small $\epsilon$.

**Algorithm 1** $\widehat{\mathcal{SI}}^*_{n,m}$ Estimation

---

Initialize NNs $f_1$ and $f_2$ and the 1-dimensional MI estimator $\widehat{I}_n$

**for** number of iterations **do**

    Sample $\psi$ and $\upsilon$ independently and uniformly on spheres $\mathbb{S}^{d_x-1}$ and $\mathbb{S}^{d_y-1}$ respectively

    Feed the random slices to $f_1$ and $f_2$: $\theta = f_1(\psi, \upsilon)$ and $\phi = f_2(\psi, \upsilon)$

    Calculate average MI over output slices: $\frac{1}{m} \sum_{i=0}^m \widehat{I}_n(\theta_i^\top X, \phi_i^\top Y)$

    Compute loss and update step for $f_1$ and $f_2$

**end for**

Feed random slices to $f_1$ and $f_2$: $\theta^* = f_1(\psi, \upsilon)$ and $\phi^* = f_2(\psi, \upsilon)$

Calculate $\widehat{\mathcal{SI}}^*_{n,m}$

---

- $\theta_0 = S\sqrt{1-\theta_1^2}, S = -1, 1$ with equal probability (Similarly for $\phi_0$).

2. To satisfy the second criterion: $\theta_0, \phi_0 \sim \mathcal{N}(0, \epsilon)$, with low probability which ensures that slices are scattered around the circle.

## D.2 Details of 'Effectiveness of $\mathcal{SI}^*$ as a Dependence Measure'

For each geometry, we generate 1000 datasets each with $10^4$ samples, where 500 sets depict the geometries with noise (positive samples)[5], and the other 500 sets are noise, i.e. statistically independent and uniformly-sampled data points (negative samples). Each measure of dependence is estimated for each set, the test thresholds the dependence measure, with independence declared when the dependence measure is below the threshold. Rather than choosing a single threshold, we report the Area Under the ROC Curve, a standard way to illustrate the performance of detectors. The ROC curve is found using the typical method: by varying over all possible thresholds and plotting the probabilities of both types of error. Importantly, both $\mathcal{SI}$ and $\mathcal{SI}^*$ were computed using $10^3$ slices. Also, error bars are too small to see due to the 100 random runs of each geometry in the experiment. Now, we list the geometries used in our experiments: (The matrices $\boldsymbol{P}, P_1, P_2, \Lambda$ are realized at the beginning of each iteration)

**Linear/parabolic/sinusoidal:** $\boldsymbol{P} \in \{0,1\}^{d\times d}, \boldsymbol{P}\boldsymbol{P}^\top = I_d, X, Z \sim \boldsymbol{\gamma}(\mathbb{S}^{d-1}), Y = f(\boldsymbol{P}X) + Z$, where $f$ is applied element-wise ($\boldsymbol{P}$ is called permutation and deletion matrix).

**Common signal:** $P_1, P_2 \in \mathbb{R}^{d\times k}, X = P_1 V + Z_1$ and $Y = P_2 V + Z_2$, $V \sim \mathcal{N}(0, I_k)$, and $Z_1, Z_2 \sim \mathcal{N}(0, I_d)$.

**Elliptical distribution:** $\Lambda \in \mathbb{R}^{d\times d}, X, Z \sim \boldsymbol{\gamma}(\mathbb{S}^{d-1}), Y = \Lambda X + Z$

**Harmonic surfaces:**

$$X, Z \sim \boldsymbol{\gamma}(\mathbb{S}^2), Y = Z + \begin{bmatrix} Y_1 \\ Y_2 \\ Y_3 \end{bmatrix} = Z + T_{\text{SH1}}(X) = Z + \begin{bmatrix} \left(\frac{X_1^2 - X_2^2}{X_1^2 + X_2^2}\right)(1 - X_3^2)^2 X_1 \\ \left(\frac{X_1^2 - X_2^2}{X_1^2 + X_2^2}\right)\sqrt{(1 - X_3^2)^3}\sqrt{\frac{1 + \sqrt{1 - X_3^2}}{2}} X_2 \\ \left(\frac{X_1^2 - X_2^2}{X_1^2 + X_2^2}\right)(1 - X_3^2)^2 X_3 \end{bmatrix}$$

$$X, Z \sim \boldsymbol{\gamma}(\mathbb{S}^2), Y = Z + \begin{bmatrix} Y_1 \\ Y_2 \\ Y_3 \end{bmatrix} = Z + T_{\text{SH2}}(X) = Z + \begin{bmatrix} \left(\frac{X_1^2 - X_2^2}{X_1^2 + X_2^2}\right)(1 - 4X_3^2(1 - X_3)^2)^2 X_1 \\ \left(\frac{X_1^2 - X_2^2}{X_1^2 + X_2^2}\right)(1 - 4X_3^2(1 - X_3)^2)^2 X_2 \\ \left(\frac{X_1^2 - X_2^2}{X_1^2 + X_2^2}\right)(1 - 4X_3^2(1 - X_3)^2)^2 X_3 \end{bmatrix}$$

## D.3 Details of Representation Learning & Reinforcement Learning

The discriminator $T$'s architecture is:

$$\text{Input} \rightarrow FC512 \rightarrow ReLU \rightarrow FC512 \rightarrow ReLU \rightarrow \text{Output} \in \mathbb{R}$$

---

[5]Given $P_{X,Y} \in \mathcal{P}(\mathbb{R}^d \times \mathbb{R}^d)$ with a specified geometry, we obtain the positive samples from the distribution $P_{X,Y} * \mathcal{N}(0, \epsilon I_{2d})$, where $\epsilon = 0.1$ and $*$ is the convolution operator. Note that noise ratio $\times$ number of samples = number of noisy data points in the positive samples set.

For $f_1, f_2$, we use a two-layer perception with normalized output:

$$\text{Input} \rightarrow FC400 \rightarrow ReLU \rightarrow FC300 \rightarrow ReLU \rightarrow \text{Output}$$

All networks use the Adam optimizer with lr=0.005, betas=(0.5, 0.999), to update their parameters. The number of gradient updates to find $f_1^*, f_2^*$ is constant throughout the experiments and is equal to 10. We ran our experiments on the STL-10 dataset (Coates et al., 2011) that uses $(96 \times 96)$ images and provides a labeled to unlabeled data ratio of one to 200 per class. As mentioned, we replicate the settings of Hjelm et al. (2019): "*data augmentation* by taking random $64 \times 64$ crops while flipping horizontally during unsupervised learning". For the encoder, we used an Alexnet (Krizhevsky et al., 2012) architecture which can be found in Donahue et al. (2016); two hidden layers with 4096 units, ReLU activations and batch norm (Ioffe and Szegedy, 2015) on every hidden layer.

For the Reinforcement Learning experiment, we use state-of-the-art SAC (Haarnoja et al., 2018) PPO (Schulman et al., 2017) and TD3 (Fujimoto et al., 2018) learning models as baselines. For the evaluation of $\mathcal{SI}^*$, We augment the reward-maximizing objective of PPO with the adapted forward information objective Equation (8) which uses $\mathcal{SI}^*$ as a dependence measure. We further augment with SI and MI versions of the same objective. Results were reported on three challenging Gym environments using the MuJoco physics engine (Todorov et al., 2012).

Throughout our experiments, we tuned the hyperparameters by performing a grid search. We found $\lambda_1 = \lambda_2 = 0.2$ for the Ant-v3 and the Hopper-v3 tasks, $\lambda_1 = \lambda_2 = 0.1$ for the Humanoid-v3 task. Furthermore, $\omega_X = \omega_Y = \pi/4, \rho = 10$ across all experiments.

## E   On Generalized Slicing Optimality

Goldfeld et al. (2022) discussed an interesting generalization of $\mathcal{SI}$ where instead of sampling slicing vectors from a sphere, one can sample matrices from a Stiefel manifold with an appropriate dimension. In what follows, we generalize our measure, $\mathcal{SI}^*$, similarly and leave the exploration of the result to future works:

**Definition 1** (Generalized Slicing Optimality). *Let $k \in \mathbb{N}$. Define the collection of probability measures $\mathcal{M}_{d,\omega} = \{\mu : \mu \in \mathcal{P}(\mathcal{V}_{k,d}), \mathbb{E}_{A,B\sim\mu}\left[\arccos\left|\operatorname{tr}(A^\top B)/k\right|\right] \geq \omega\}$, where $\mathcal{V}_{k,d} \stackrel{def}{=} \{X \in \mathbb{R}^{d\times k}, X^\top X = I_k\}$ is the Stiefel manifold. The Generalized form of slicing optimality is:*

$$\mathcal{GSI}_k^*(X;Y) = \sup \oint_{\mathcal{V}_{k,d_x} \times \mathcal{V}_{k,d_y}} \mathcal{I}(U^\top X; V^\top Y) d\sigma(U,V),$$

*where the supremum is taken over $\sigma \in \mathcal{P}(\mathcal{V}_{k,d_x} \times \mathcal{V}_{k,d_y})$ with the constraint that its marginals should lie in $\mathcal{M}_{d_x,\omega_X}$ and $\mathcal{M}_{d_y,\omega_Y}$, respectively.*

$\mathcal{GSI}_k^*$ forms a general class of sliced information measures (if $k = 1$, $\mathcal{GSI}_k^* = \mathcal{SI}^*$).