# OpenReview forum: "On Slicing Optimality for Mutual Information"
_NeurIPS.cc/2023/Conference — NeurIPS 2023 poster_

### Official Review · Reviewer_8Sok · 2023-06-27

**Soundness:** 3 good
**Presentation:** 4 excellent
**Contribution:** 2 fair
**Rating:** 7
**Confidence:** 3

**Summary:**

The paper estalishes a new quantity for measuring the dependence between random variables. The definition is similar to mutual information and sliced mutual information. As (for rather obvious reasons) it takes values between these two existing measures, it has some desireable properties. A variational representation further makes it better compatible with neural estimators.

**Strengths:**

The paper is well written, provides good references and approaches the challenging gap between theory of information measures and their estimation in practice.

**Weaknesses:**

It is difficult to see the benefit of the specific definition (2). There is no convincing argument why (2) is better than (1). Also an optimization over a set of prabability measures appears to be much more difficult to approximate than the comparatively "simple" integration over the uniform measure.
Many details are in appendices that I unfortunately have no time to read in the limited time given for the review process of many papers.

**Questions:**

Are there methods that try to approximate (1) by other means than uniform sampling? There are many methods for numerical integration that might also use the fact that not all directions are equally useful in approximating the integral.
Why is it necessary to change the definition of (1) and not simply try to use a better method to approximate (1) numerically?
Does there exist a one-dimensional MI estimator with finite \delta(n)? If not, can the supremum in the characterizing property of \delta(n) be restricted to a class where it is finite and all further results hold for this class?

**Limitations:**

Limitations were not addressed in the paper but discussed in an appendix.

---

> ### Author Rebuttal · Authors · 2023-08-10
>
> We thank you for the positive review and helpful comments. Below, we address your comments and questions.
>
> **Why (2) is better than (1)?** In low rank settings, Sliced mutual information (1), $\mathcal{SI}$ [1] performs quite well despite the fact that our method (2) still surpasses $\mathcal{SI}$ in performance (e.g: Fig.2 common signal). However, as the structure becomes more complex and the dimensions increase, $\mathcal{SI}$'s performance worsens (e.g: Fig.2 Elliptical and Fig.3 Elliptical) but $\mathcal{SI}^*$ maintains the good performance. Based on this behavior of $\mathcal{SI}$, it was noted that the problem was due to the randomness of slices as they carry ``little information". Indeed, since the slicing technique essentially averages MI terms, it can offer limited improvements over the classical MI, which is the case for $\mathcal{SI}$ and that is why we chose to not numerically approximate it. To resolve this dilemma, our work proposes a novel measure (Eq.2) that finds distributions of the slices that carry as much information as possible, hence less noise and better gains.
>
> Now despite the fact that an optimization over a set of probability measures can be more ``difficult" than a simple integration over the uniform distribution, it doesn't cancel the need for new measures of dependence that don't suffer from the problem above. Indeed, in addition to overcoming this issue, our measure offers several advantages. These include finding an optimal distribution therefore reducing the number of needed slices to the minimum, the ability to uncover complex relationships that other measures fail to reveal, and a significantly improved performance especially in higher dimensions. Moreover, our empirical findings, detailed in the paper, demonstrate that the computational complexity of $\mathcal{SI}^*$ is of the same order as that of  $\mathcal{SI}$.
>
> As for the MI estimator, indeed there exists one with $\delta(n)\leq C\big((n\log n)^{-\frac{s}{s+2}} \log n ^{1-\frac{2}{p(s+2)}} +n^{-1/2}\big)$ where $s,p: 0<s\leq2\leq p,$ and $ C>0$ (line 230). This term is finite and there is no further need to restrict it to a specific class.
>
> [1] Goldfeld, Z. and Greenewald, K. Sliced mutual information: A scalable measure of statistical dependence, NeurIPS 2021.

---

> > ### Comment · Reviewer_8Sok · 2023-08-16
> >
> > Thank you for your detailed response. I increased my score as I before missed the reference in line 230.

---

### Official Review · Reviewer_8cdY · 2023-07-06

**Soundness:** 2 fair
**Presentation:** 3 good
**Contribution:** 2 fair
**Rating:** 5
**Confidence:** 4

**Summary:**

Recently, sliced mutual information ($\mathcal{SI}$) has been proposed as a statistical dependence measure that is scalable to high dimensions (Goldfeld and Greenewald, 2021). However, due to the use of uniform sampling for the slicing directions, $\mathcal{SI}$ generally requires many slices for accurate estimation. This is because it doesn’t prioritize informative features about the variables and introduces noisy slices by treating both relevant and irrelevant regions of the variable space equally. This paper aims to improve $\mathcal{SI}$ by finding an optimal distribution of slicing directions that satisfies two criteria: 1) slices are distributed over maximally informative regions, and 2) slices are scattered over the unit sphere. This new dependence measure is called optimal sliced mutual information, $\mathcal{SI}^*$. The two criteria are then formalized as a regularized optimization problem in the definition of $\mathcal{SI}^*$.

The contributions of this paper are as follows: (i) The authors introduce an improved version of $\mathcal{SI}$ known as $\mathcal{SI}^*$ and discuss its theoretical properties. In particular, they show that an optimal slicing policy exists, allowing them to treat the regularized optimization problem as a reward-maximizing optimal transport problem. (ii) The authors show that the convergence rate of the optimal estimator of  $\mathcal{SI}^*$ depends on the dimensions only up to a constant factor. In addition, they develop a neural estimator for $\mathcal{SI}^*$ which can be used in various machine learning problems. (iii) The authors perform several experiments to show the effectiveness and applications of $\mathcal{SI}^*$. In particular, they show that: (a) $\mathcal{SI}^*$ outperforms state-of-the-art measures in discerning structure from noise, (b) $\mathcal{SI}^*$ is more sample- and slice-efficient compared to $\mathcal{SI}$, and (c) $\mathcal{SI}^*$ performs well on representation and reinforcement learning problems.

**Strengths:**

The paper improves $\mathcal{SI}$ by challenging the uniform sampling assumption in the original definition. The paper is original as it proposes a novel, improved measure of $\mathcal{SI}$ that can capture statistical dependency better by retaining as much relevant information as possible. The theoretical result on the existence of the “optimal” slicing policy is interesting and non-trivial. The experiments clearly showcase the effectiveness of $\mathcal{SI}^*$ as a dependence measure and how it can be applied to various machine learning tasks. The paper shows how $\mathcal{SI}^*$ is an effective statistical dependence measure that works well in high dimensions. As many machine learning tasks involve high-dimensional variables, this measure will prove useful in practice. Overall, the paper is well-structured and was an interesting read.

**Weaknesses:**

While the paper has some interesting points, there are several weaknesses that should be addressed to make the paper more readable.

1. While the definition is $\mathcal{SI}^*$ is intriguing, the justification for the two criteria for “optimality” that motivated the definition is inadequate. (i) Why is it relevant or important to consider the maximally informative regions? (ii) If we accept that the maximally informative regions are important, why do we need criterion 2? The authors should explain more clearly why diversity of slices is important when measuring dependence between random variables.

1. From the definition of $\mathcal{SI}^*$, wouldn’t it be better to call it something like “maximum $\mathcal{SI}$”?

1. In Figure 2, the experiments are done for variables in $\mathbb{R}^3$. However, $\mathcal{SI}^*$, like $\mathcal{SI}$, should continue to be scalable to high dimensions. Thus, the experiments should be done in higher dimensions (> 100), so we can test if the three neural networks involved in the estimation of $\mathcal{SI}^*$ scale well with larger dimensions of inputs.

1. Parts of the paper were hard to understand and could have been explained more clearly. (i) For example, the terms in the neural network estimator (Section 4) need more details and better explanation. (ii) In the motivating example (Section 3.1), it is not clear what the AUC of the ROC refers to or how it is computed in this case. The Appendix provides some details for another experiment but it is not clear if that applies here. Also, it seems that Fig. 1 is not an ROC AUC curve despite the caption stating as such. (iii) The setup and assumptions for the experiments require more details beyond that given in the paper and appendix.

1. Since one claim in the paper is better sample and slice efficiency of $\mathcal{SI}^*$, is it possible to provide theoretical results along this direction? The results in the paper seem to indicate that $\mathcal{SI}^*$ has a similar convergence rate as $\mathcal{SI}$, thus reducing its significance.

1. Another claim in the paper is the potential of $\mathcal{SI}^*$ to bring the approach to more complex machine learning scenarios. In this context, more experimental evaluation is necessary for both the representation learning and reinforcement learning scenarios. The current evaluation (e.g., Table 1) is not to the state of the art, in fact, a shallow vanilla CNN without data augmentation has 86% accuracy on CIFAR-10. This makes it difficult to interpret the significance of the results.

**Questions:**

In addition to the list of weaknesses above, below are some questions and general suggestions for the authors to address:

1. In Goldfeld and Greenewald’s paper, they show that $\mathcal{SI}$ can increase with more processing of the random variables. Can $\mathcal{SI}^*$ increase with more processing as well?
Can we apply chain rule to $\mathcal{SI}^*$?

1. In Appendix B.1, the authors computed the RMSE between the ground truth and empirical values of MI and $\mathcal{SI}^*$ as a function of the number of samples and slices (where applicable). Please include the relevant results for $\mathcal{SI}$ as well.
1. In Appendix B.1, the authors computed the RMSE between the ground truth $\mathcal{SI}^*$ and the estimated $\mathcal{SI}^*$. It is not clear how the ground truth $\mathcal{SI}^*$ is computed in this case. In Goldfeld and Greenewald’s paper, they provided an expression for Gaussian $\mathcal{SI}$ (Example 1). Is it possible to come up with similar expression for $\mathcal{SI}^*$?

1. In Appendix C, it seems that the algorithm is lacking in detail and clarity. For example, the authors should make it clear that $m$ number of random slices are being sampled. It is also not clear in the algorithm what the loss functions used for training the various neural networks are. Overall, the description of the algorithm seems a bit scattered and could be significantly made clearer.

1. Goldfeld and Greenewald have considered extending $\mathcal{SI}$ to $k-\mathcal{SI}$ where instead of projecting to 1 dimension, the random variables are projected to $k$ dimensions. Can $\mathcal{SI}^*$ be easily extended to the $k$ dimensions case?

1. Minor comments: (i) Fix caption of Fig. 1 to not refer to the ROC curve. (ii) In Appendix D.1, $\phi$ should be $\varphi$.

**Limitations:**

The limitations are discussed in Appendix E.

---

> ### Author Rebuttal · Authors · 2023-08-10
>
> Many thanks for the constructive feedback. We address all your concerns below.
> 1. Reviewer posed a good question. In lines 140-148, we discussed the reasoning behind the definition of eq 2 and continued to offer evidence that shows the effectiveness of our criteria. We further discuss the question of "why" here and we will incorporate it in the paper.
>
>    To answer (i), let's note the following: in low rank settings, Sliced mutual information, $\mathcal{SI}$ [1] performs quite well despite the fact that our method still surpasses $\mathcal{SI}$ in performance (e.g: Fig.2 common signal). However, as the structure becomes more complex and the dimensions increase, $\mathcal{SI}$'s performance worsens (e.g: Fig.2 Elliptical and Fig.3 Elliptical) but our method maintains the good performance. Based on this behavior of $\mathcal{SI}$, it was noted that the problem was due to the randomness of slices as they carry ``little information". Indeed, since the slicing technique essentially averages MI terms, it can offer limited improvements over MI, which is the case for $\mathcal{SI}$. Thus, in our paper, we overcome this dilemma by searching for distributions of the slices that carry as much information as possible (i.e. concentrating the slices in the maximally informative regions) to avoid noise, information redundancy, and little gains.
>
>    As for (ii), a main technical benefit thanks to the second criterion is that the the slicing distributions cannot collapse into the Dirac measure. The latter along with the uniform distribution are too restricted since in the abundance of complex data, viewing all directions equally or considering only one direction is inefficient. It is thus intuitive to consider areas around important slicing directions. But restricting ourselves to only one important area is not good since under the slicing in one area, the projected random variables will have close mutual information values and hence some redundancy, not to mention that other areas can carry significant information as well. In short, the second criterion prevents the collapse into Dirac measure and encourages efficient "exploration" of the space of slicing directions to ensure not missing any informative regions.
> 2. Maximum $\mathcal{SI}$, which was discussed in [1], might be misinterpreted as a maximum over slices rather than over distributions that gives maximum information.
> 3. We added 3 experiments with dimensions 100 and 200 on 3 complex geometries. See the pdf. However, the experiment in Figure 2 is a proof of concept and is intended to showcase the efficiency of $\mathcal{SI}^*$ in capturing complex relationships between noisy variables. Later experiments support the authors claim that $\mathcal{SI}^*$ scales to higher dimensions. Specifically, in Figure 3 we study variables with varying dimensions and in the reinforcement learning setting we apply our dependence measure to variables of dimensions up to [378, 395]=[observation dim, observation dim+action dim].
> 4. (i) Neural estimator details: In corollary 2, we derived a variational representation of $\mathcal{SI}^*$, which we used to formulate the empirical objective of the neural estimator. The notations and other details carry over.
>     Specifically, Eq.6 is nothing but an empirical version of Eq.3: we learn $f_1,f_2$ to push forward the uniform measure to the optimal slicing measure.
>     The last two terms in Eq.6 do account for the $\arccos|.|$ constraint in definition 1 (aka the second criterion). Moreover, $\lambda_1$ and $\lambda_2$ have the role of tuning the last two terms. Lastly, $\omega_X,\omega_Y$ are hyperparameters defined in Definition 1. We provided an ablation study for them in App.B2.
>
>    (ii) The same settings that apply to other ROC experiments apply in the motivational example. We calculate the AUC of the ROC to evaluate the discriminative ability of $\mathcal{SI}$ with a uniform distribution of slices (Green) and our modified version of the slicing distribution (Red). If the source of confusion is the blue dashed line, that is because it shows the AUC of MI, which doesn't change as MI is independent of the number of slices.
> 5. Since the reviewer is referring to the convergence rate in theorem 4, then in fact, having a similar convergence rate to $\mathcal{SI}$ is itself a breakthrough because having a novel measure with a superior performance that retains the convergence properties of $\mathcal{SI}$ and depends on the dimensions up to a constant factor is quite significant and proved useful in numerous applications.
>
>     The theoretical analysis of sample and slicing sizes effects on dependence detection power is still missing from the literature as a whole, that is because we do not have the mathematical tools to quantify the quality of dependence measures through theoretical means. We thus content ourselves by proving our claims about the efficiency of our measure through extensive experiments.
> 6. Within the time limit, we provide an extra experiment on the Half-Cheetah environment in the pdf. However please note that in our experiments on representation learning and reinforcement learning, we aim to show  scalability and compatibility of $\mathcal{SI}^*$ with machine learning models in addition to **improvement** in the performance of a given algorithm upon adopting our measure. So the existence of algorithms with better outcomes does not reduce the significance of the contributions made in the paper. With that being said, it is hard to overstate the 25\% improvement of return in our reinforcement learning experiments, which to our best knowledge is a new record.
>
> We answer your questions in the common rebuttal above.
>
> We believe we have addressed all your concerns and are looking forward to you raising the score accordingly.
>
> [1] Goldfeld, Z. and Greenewald, K. Sliced mutual information: A scalable measure of statistical
> dependence, NeurIPS 2021.
>
> [2] Kim et. al, EMI: Exploration with Mutual Information, ICML 2019.

---

> > ### Comment · Reviewer_8cdY · 2023-08-21
> >
> > I would like to thank the authors for addressing my questions and providing clarifications. Thank you also for the new experiments. Please include the new experiments with some discussion and the $k$-dimension definition to the appendix. I have increased my score accordingly. All the best.

---

### Official Review · Reviewer_k46z · 2023-07-07

**Soundness:** 3 good
**Presentation:** 3 good
**Contribution:** 3 good
**Rating:** 6
**Confidence:** 3

**Summary:**

This work based on existing ideas of providing a tractable independence metric for high dimensional random variables, the proposed metric named "Optimal sliced mutual information" (SI*) improves upon existing method of uniform slicing to an adaptive optimal slicing with scalability design in mind. It has been shown that the proposed method has desirable properties to be a dependency metric. The estimator is shown to be faster converging than previous MI both theoretically and empirically. In addition, this method is applied on several machine learning tasks to show its effectiveness.

**Strengths:**

This work discovers weakness in previous work in terms of slicing efficiency. The idea is well motivated and explained well in introductory sections. The properties of the proposed metric is well explained with easy to understand examples and visualization. Examples in the empirical study showcases various in field applications which implies wide impact upon adoption.

**Weaknesses:**

The scalability of the NN estimator of SI* is only studied at slicing size level. A remark about the scalability with $d_x, d_y$ similar to convergence analysis can be beneficial.

**Questions:**


There are multiple places where SI and $\mathcal{SI}$ are used interchangeably. e.g. l.50 l.282 Table 1, l.331 and more.
The provided implementation is short scaffolding code and missing many important pieces which prevents further cross-check of the implementation against the claim in the paper.

**Limitations:**

Limitations are well addressed.

---

> ### Author Rebuttal · Authors · 2023-08-10
>
> We thank you for the positive review and helpful comments.
>
> We conduct an experiment on the NN estimator at the sample size level while varying $d_x,d_y$. We have $X,Y\in\mathbb{R}^d$ with linear dependence. The results are in the attached pdf above. One can tell, similarly to the convergence analysis in the appendix, that there is no significant difference in the outcomes as the dimensions vary. Lastly, we provided the implementation details in addition to an example code to allow reproducibility. We will publish the code base for all the experiments with the camera-ready version.

---

> > ### Comment · Reviewer_k46z · 2023-08-12
> >
> > Thanks for the response. I have also carefully gone over other reviewers opinions available.
> >
> > A follow up Q:
> >
> > Line 225 implies when $d_x, d_y$ increases, $U$ decrease, in other words the estimation error upper bound should be smaller. But the three lines for $d=10,50,100$ show a trend of increasing rmse.

---

> > > ### Author Response · Authors · 2023-08-12
> > >
> > > Thanks again for your time, and this excellent observation.
> > >
> > > In line 225, the term $U$ is involved in the error bound of the estimator in Eq.5 that is based on a 1-dimensional MI estimator and NOT the neural estimator (Eq.6). So the plots in the rebuttal file need not show a trend of decreasing RMSE with dimensions. However, in the appendix Figure 5(c) where we study the estimator that has $U$ in its error upper bound, you can see that the RMSE when dimensions are 100 is less than the RMSE when dimensions are 50. On that note,  for some low-dimensional $X,Y$, the effect of $U$ might not be apparent, that's because $U$ is only involved in the upper bound of the uniform error. The impact of $U$ should become much more noticeable as dimensions get high.

---

### Official Review · Reviewer_w6uv · 2023-07-10

**Soundness:** 3 good
**Presentation:** 4 excellent
**Contribution:** 3 good
**Rating:** 6
**Confidence:** 3

**Summary:**

This paper proposes a new dependence measure based on the sliced mutual information with both the theoretical analysis and the modern machine learning experiments. This measure is not strictly the mutual information but still could capture the correlation between two distributions.
The experiments are done in several different domains in eluding image classification, reinforcement learning. Their main idea is to replace the original uniformly sampled slices to the one whose directions could be diversified over the whole sphere as well as mainly concentrates on the interested area.


**Strengths:**

The article is well written, starting from an easy understanding toy example, followed by the motivation and explanation. The authors also clarify several important concepts that the reviewer will raise during the reading. The theoretical analysis is detailed and clearly written.
- The observation of the the uniform distribution in sliced mutual information is insightful
- Extensive experiments are done in different domains

**Weaknesses:**

Have two minor concern and hope to hear from the authors,
- In the theoretical part, the authors prove the existence of the optimal solution using the optimal transport's view. However, during the neural estimation, the problem is simplified to gradient descend with Lagrange constraints. The hyper parameters listed in equation (6) are not clearly described.
 - The efficiency is not as good as the previous methods (ten times comparing with SI. Which factors cause the computation speed to be slow?

**Questions:**

- Could you please explain what is the definition of AUC that shown in this paper. Does the evaluation treat as a binary classification of dependence and independence? Since this metrics are used in at least two experiments, could you please put more words on this part?
- Many different mutual information estimator are mentioned in the paper, including the discriminator-based form and variational representation learning. There are also many other mutual information estimator, why did the author choice MINE[1]?
- What neural estimator do you use in Table 1 DIM(MI)?
- How do you choose $\lambda_1$, $\lambda_2$, is there any ablation study on this part?



[1] Mutual Information Neural Estimation, 2021

**Limitations:**

No negative societal impact.

---

> ### Author Rebuttal · Authors · 2023-08-10
>
> We thank you for the positive review and helpful comments. Below, we address your concerns and questions in order.
> 1. Followed from Corollary 3, the Lagrangian dual form in Eq.6 is completely equivalent to the original definition of $\mathcal{SI}^*$. The last two terms in Eq.6 account for the $\arccos|.|$ constraint in definition 1 (aka the second criterion) and $\lambda_1$ and $\lambda_2$ have the role of tuning the last two terms. An important note can be made about $\lambda_1, \lambda_2$: in high dimensions, we set $\lambda_i$ to be smaller than in the case of low dimensions, because sampled high dimensional vectors are almost orthogonal which automatically satisfies the $\arccos|.|$ constraint. In the pdf, we added the ablation for $\lambda_1,\lambda_2$ which hints a that observation.
>
>     The other hyperparameters used are $\omega_X, \omega_Y$, which have been defined in Definition 1, and we studied their impact in App B2. Lastly, $\rho$ is used to clip the exponential term in Eq 6 to account for the possible high-variance issues as discussed in [1].
> 2. $\mathcal{SI}^*$ and $\mathcal{SI}$ share a computational complexity of the same order as shown in Figure 7; in some cases, $\mathcal{SI}$ performs up to $1.661$ times better than $\mathcal{SI}^*$. This difference is insignificant especially given the superiority of $\mathcal{SI}^*$ over all other baselines and its ability to uncover more complex relationships. *Which factors affect the computation speed?* Mainly the training of three neural networks.
> 3. For each experiment, we generate $1000$ sets each containing $10^4$ samples. $500$ of these samples represent geometries with noise (positive samples), and the other $500$ consist of statistically independent and uniformly-sampled data points (negative samples). We calculate the intended measure of dependence for each set and then apply a test by thresholding the dependence measure; should the measure value be lower than the threshold, we conclude that there is independence. Instead of selecting a single threshold, we find the ROC curve through different thresholds, then the area under the curve (AUC) is computed to evaluate performance. Of course, the higher the AUC is, the better the measure is at discerning structure from independently sampled data. We will add these more details to the paper.
> 4. Given the variational representation of $\mathcal{SI}^*$, it is very flexible and straightforward to parameterize $T$ in Eq.3 similarly to MINE [2].
> 5. We replicated the setting in [3], namely, using the MINE estimator.
>
> [1] Song, J. and Ermon, S. Understanding the limitations of variational mutual information estimators. ICLR 2020.
>
> [2] Belghazi et al. Mutual Information Neural Estimation. ICML 2018.
>
> [3] Hjelm et al. Learning deep representations by mutual information estimation and maximization. ICLR 2019.

---

### Author Rebuttal · Authors · 2023-08-10

All reviews and feedback are appreciated, and many suggestions will be incorporated in the paper.

Attached you can find a file containing new experiments suggested by the reviewers.

* Reviewer 8cdY: (1) AUROC with high dimensions (>100); Figure 1. (2) An RL experiment on Half-Cheetah-v3; Figure 2(c). (3) Include $\mathcal{SI}$ convergence results; Figure 2(a).

* Reviewer k46z: Scalability of the NN estimator with varying $d_x,d_y$ at the sample size level; Figure 2(b).

* Reviewer w6uv: Simple ablation study on $\lambda_1, \lambda_2$; Table 1.


In this remaining space below, we answer the questions raised by **reviewer 8cdY** in order.
1. $\mathcal{SI}^*$ grows as a result of some deterministic transformations. In fact, for $\omega_X,\omega_Y\geq \pi/4$, $\sup_{\alpha,\beta} \mathcal{SI}^*(nn_\alpha(X);nn_\beta(Y)) = \sup_{\theta,\varphi}\mathcal{I}(\theta^t X; \varphi^t Y)$, where $nn_\alpha(.)$ is a shallow neural network with parameters $\alpha.$ The proof follows directly from Theorem 3.
    As mentioned in the limitations section, the study of the multivariate settings (including the chain rule) is encouraged for future works.
2. We included the relevant results for $\mathcal{SI}$ in the attached file.
3. The ground truth value is computed by taking a high number of slices and samples chosen to be sufficiently large to guarantee the convergence of $\mathcal{SI}^*$. Also, both $\mathcal{SI}^*$ and $\mathcal{SI}$ don't have closed formulas for normal distributions because the integration over slices cannot be computed in closed form.
4. Regarding the pseudo-code, we calculate the term in Eq.5 (main paper), which is the objective function, and use stochastic gradient ascent to update $f_1,f_2$. We will incorporate those details in the chart to make the algorithm description clearer.
5. Can $\mathcal{SI}^*$ be easily extended to the $k$ dimensions case? Great question! The answer is yes, and we have considered adding that to the paper but we were concerned it might shift the attention of the reader. The definition is included in the attached file. It retains the dependence properties mentioned in theorem 2. We will add it to the appendix if the reviewer deems it necessary.

---

### Decision · Program_Chairs · 2023-09-21

**Decision:**

Accept (poster)

**Comment:**

This paper proposes a new independence measure based on slicing technique for high-dimensional variables. The key idea is to use the slicing techniques mainly studied in optimal transport community for mutual information like measure. The proposed measure is well supported by theory and experiments. Overall, the idea is interesting and the reviewers agree to accept the paper. So, I also vote for acceptance.

Although the proposed method is interesting and new, it seems some important previous work is missing. So, in the camera-ready, please include the following papers and discuss their relation to your methods. The definition (Optimal sliced mutual information) seems very related to information maximization-based optimal transport methods.

Liu et al., LSMI-Sinkhorn: Semi-supervised Mutual Information Estimation with Optimal Transport. ECML 2021

Chuang et al., InfoOT: Information Maximizing Optimal Transport. ICML 2023